# Kronecker Generative Models for Power Law Patterns in Real-World Hypergraphs

## ABSTRACT

Do real-world hypergraphs obey any patterns? Are power-laws fundamental in hypergraphs as they are in real-world graphs? What generator can reproduce these patterns? A hypergraph is a generalization of a conventional graph, and it consists of nodes and hyperedges, with each hyperedge joining any number of nodes. Hypergraphs are adept at representing group interactions where two or more entities interact simultaneously, such as collaborative research and group discussions.

In a wide range of real-world hypergraphs, we discover power-law or log-logistic distributions in eight structural properties. To simulate these observed patterns, we introduce HyRec, a tractable and realistic generative model leveraging the Kronecker product. We mathematically demonstrate that HyRec accurately reproduces both the patterns we observed and typical evolutionary trends found in real-world hypergraphs. To fit the parameters of HyRec to large-scale hypergraphs, we design SingFit, a fast and space-efficient algorithm successfully applied to eleven real-world hypergraphs with up to *one million* nodes and hyperedges.

This paper makes the following contributions: (a) *Discoveries*: we identify multiple patterns that real-world hypergraphs obey, (b) *Model*: we propose HyRec, a tractable and realistic model capable of reproducing real-world hypergraphs efficiently (spec., with fewer than 1,000 parameters) with the support of SingFit, and (c) *Proofs*: we prove that HyRec adheres to these patterns.

## 1 INTRODUCTION

In the real world, group interactions, such as collaborative research, co-purchases of items, and group discussions on online platforms, are prevalent. These are well-represented by *hypergraphs*, where each hyperedge indicates a group interaction as a subset of nodes of any size. Hypergraphs extend conventional graphs, overcoming their limitation of exclusively modeling pairwise interactions.

What patterns or "laws" shape the structure of real-world hypergraphs? While power-laws are fundamental in real-world graphs [15, 19], are they also prevalent in hypergraphs? To answer this, we analyze eight power-law properties across eleven real-world hypergraphs. First, we confirm the presence of power-laws in previously identified heavy-tailed distributions, specifically for node pair degrees and hyperedge intersection sizes in hypergraphs [13, 24, 27], by applying linear regression fitting on a log-log scale. We also reveal that the slopes of these linear regressions are consistent

*Conference acronym 'XX, June 03–05, 2018, Woodstock, NY*
© 2024 Copyright held by the owner/author(s). Publication rights licensed to ACM.
ACM ISBN 978-1-4503-XXXX-X/18/06...$15.00
https://doi.org/XXXXXXX.XXXXXXX

within the same domains. Then, we find that the distributions of node degrees and hyperedge sizes follow log-logistic distributions, which are mathematically closely related to power-laws. Lastly, we uncover new power-law patterns related to clustering coefficients, density, and overlap[27] in hypergraphs.

What mechanisms underlie the complex structures of real-world hypergraphs, and how can we effectively model them? Inspired by the Kronecker graph model [29], a successful generative model for conventional graphs, we introduce HyRec, a new generative model for hypergraphs. In essence, it yields an incidence matrix, indicating which nodes belong to which hyperedges, through the Kronecker power of a small-sized initiator matrix. We mathematically prove that HyRec yields five structural properties following multinomial distributions, which can mimic power-law and log-logistic distributions [5, 10, 29], and simulates evolutionary patterns of real-world hypergraphs, such as densification and shrinking diameters [24].

Since HyRec generates hypergraphs using Kronecker products of the initiator matrix, finding the most suitable initiator matrix to accurately reflect a specific real-world hypergraph is critical for the model's success. This poses significant challenges, including (C1) identifying node correspondences, (C2) ensuring differentiable generation, and (C3) maintaining computational cost feasible. To address these challenges, we propose SingFit, a fast and space-efficient fitting algorithm for HyRec. SingFit (S1) circumvents the node correspondence issue by aligning singular values, instead of hyperedge occurrences, (S2) employs Gumbel-Softmax [20] for continuous approximation of sampling to ensure differentiability, and (S3) leverages Kronecker product properties by dividing the full-matrix sampling into smaller matrices, reducing both space and time requirements.

In various real-world hypergraphs, HyRec, facilitated by SingFit, demonstrates its efficacy in two practical scenarios: (1) **Fitting**: generating hypergraphs that closely replicate real-world hypergraphs with minimal parameters, and (2) **Extrapolation**: forecasting their future growth, offering insights into potential evolutionary trends. Our contributions are as follows:

- **Discoveries**: We find out that real-world hypergraphs exhibit power-law or log-logistic distributions in various properties.
- **Model**: We propose HyRec, a tractable generative model that accurately replicates real-world hypergraph properties with a small number of parameters.
- **Proofs**: We mathematically prove that HyRec is able to replicate the discovered realistic power-law and log-logistic distributions (see Theorem 1 and 2 in Section 5).

For reproducibility, our code and data are available at [1].

## 2 RELATED WORK

### 2.1 Kronecker Graph Models

The Kronecker graph model [29] is recognized for its simplicity and theoretical depth, offering a lens to understand real network

**Table 1: Comparison of Hypergraph Generative Models.** Our HyRec is the *only* model that matches all the specs. HyperCL is labeled as '?' since only analysis of node degrees and hyperedge sizes is available for it.

| Capability | No Size Limit | Theoretical Analysis | Not Requiring Detailed Statistics | Extra-polation |
|---|---|---|---|---|
| HyperCL [27] | ✔ | ? | | |
| HyperFF [24] | ✔ | | ✔ | ✔ |
| HyperPA [13] | | | | ✔ |
| HyperLAP [27] | ✔ | | | |
| THera [21] | ✔ | | | ✔ |
| **HyRec** | ✔ | ✔ | ✔ | ✔ |

dynamics with a minimal set of parameters. It adeptly mimics real-world network characteristics, such as heavy-tailed degree, eigenvalue, and eigenvector distributions, making it a standard for benchmarks like Graph500 [36]. Due to its tractability, the Kronecker graph model has inspired detailed studies on its behavior, addressing aspects such as degree distributions [39], isolated nodes, triangles [37, 39], and network connectivity [33]. When applied to bipartite graphs, the Kronecker graph model reveals patterns like scaling laws for edge clustering coefficients [42], which are relevant to our study given the bipartite nature of hypergraphs in linking nodes to hyperedges. However, our study prioritizes unique attributes of hypergraphs, such as hyperedge intersection, extending beyond the insights provided by bipartite graph analysis.

## 2.2 Fitting Algorithms for Kronecker Graphs

Kronecker graph fitting algorithms employ varied strategies for model fitting. Maximum-likelihood methods [29] aim to match edge occurrences, aligning the output adjacency matrix with the target graph adjacency matrix. In contrast, method-of-moments estimators [17] focus on matching the counts of edges, triangles, 2-stars, and 3-stars. A variants using empirically estimated moments has also been developed [35]. HyperKron [14] targets the matching of triangle motifs in graphs using 3D tensor Kronecker products. While it applies to uniform hypergraphs, where all hyperedges are of equal size, it is not directly suitable for real-world hypergraphs characterized by diverse hyperedge sizes.

## 2.3 Properties of Real-World Hypergraphs

Real-world hypergraphs exhibit heavy-tailed distributions for various properties, including node degrees [13], hyperedge sizes [24], intersection sizes [24], singular values of incidence matrices [24], and node-pair degrees [27]. Moreover, hyperedges in real-world hypergraphs tend to overlap more significantly [27] with higher transitivity [21], compared to those in random counterparts. Dynamics of time-evolving hypergraphs, including diminishing overlaps, densification, and shrinking diameters [24], have also been explored. Specifically, prior studies have examined dynamics regarding repetition [4], recency [4], burstiness [6], persistence [6, 9], ego-network structures [11], and triadic closures among nodes or hyperedges [3, 28]. For a comprehensive overview of patterns in real-world hypergraphs, see the survey [26].

## 2.4 Hypergraph Generators for Realistic Structural Patterns

Recent efforts to reproduce the structural characteristics of real-world hypergraphs through generative models focus on various

**Table 2: Summary of Real-world Hypergraphs.**

| Dataset | # Nodes | # Hyperedges | Max. Degree | Max. Size |
|---|---|---|---|---|
| email-Enron | 143 | 10,885 | 1,327 | 37 |
| email-Eu | 1,005 | 25,148 | 8,664 | 40 |
| contact-primary | 242 | 106,879 | 2,234 | 5 |
| contact-high | 327 | 172,035 | 4,495 | 5 |
| NDC-classes | 1,161 | 49,726 | 5,358 | 39 |
| NDC-substances | 5,556 | 112,919 | 6,693 | 187 |
| tags-ubuntu | 3,029 | 271,233 | 21,004 | 5 |
| tags-math | 1,629 | 822,059 | 71,046 | 5 |
| threads-ubuntu | 125,602 | 192,947 | 2,332 | 14 |
| threads-math | 176,445 | 719,792 | 12,511 | 21 |
| coauth-geology | 1,261,129 | 1,591,166 | 1,153 | 284 |

aspects, including node subset connectivity [13], modularity [16], heavy-tailed distributions [24], hyperedge overlaps [27], and transitivity [21]. Most of them, however, require detailed hypergraph statistics, such as hyperedge size distributions, as inputs for accurate reproduction (see Table 1). While HyperFF [24] models temporal dynamics without requiring any statistics, it struggles to accurately model hyperedge overlaps [23]. Moreover, the complexity of these models makes theoretical analysis challenging. Although the HMPA model [16] offers a theoretical basis, its extensive parameter set makes fitting to specific real-world hypergraphs challenging.

## 3 PRELIMINARIES AND DATASETS

In this section, we introduce the preliminaries and real-world hypergraph datasets used throughout the paper.

### 3.1 Hypergraph and Incidence Matrix

A **hypergraph** $\mathcal{G} = (\mathcal{V}, \mathcal{E})$ consists of a set of nodes $\mathcal{V} = \{v_1, \ldots, v_N\}$ and a set of hyperedges $\mathcal{E} = \{e_1, \ldots, e_M\} \subseteq 2^{\mathcal{V}}$. Each hyperedge $e \in \mathcal{E}$ is a non-empty subset of $\mathcal{V}$. The **degree** of a node $v$, denoted by $d_v$, is defined as the number of hyperedges containing $v$. A hypergraph can also be expressed by an **incidence matrix** $I(\mathcal{G}) \in \{0, 1\}^{N \times M}$, where each $(i, j)$-th entry $g_{i,j}$ of $I(\mathcal{G})$ is 1 if and only if the hyperedge $e_j$ contains the node $v_i$. A **path** in a hypergraph is defined as a sequence of hyperedges $(e_{p_1}, \cdots, e_{p_L})$ where $e_{p_i} \cap e_{p_{i+1}} \neq \varnothing$ for every $i \in \{1, \ldots, L-1\}$. The distance between two nodes $(v_i, v_j)$ is defined as the length of a shortest path $(e_{p_1}, \cdots, e_{p_L})$ where $v_i \in e_{p_1}$ and $v_j \in e_{p_L}$. The **diameter** of the hypergraph is the maximum distance between any pairs of nodes; the **effective diameter** [31] is the minimum distance within which 90% or more of node pairs are reachable.

### 3.2 Kronecker Product and Power

Given two matrices $\mathbf{A} \in \mathbb{R}^{N \times M}$ and $\mathbf{B} \in \mathbb{R}^{P \times Q}$, the kronecker product $\mathbf{A} \otimes \mathbf{B} \in \mathbb{R}^{NP \times MQ}$ is a matrix formed by multiplying $\mathbf{B}$ by each element of $\mathbf{A}$, i.e.,

$$\mathbf{A} \otimes \mathbf{B} := \begin{bmatrix} a_{11}\mathbf{B} & \cdots & a_{1M}\mathbf{B} \\ \vdots & \ddots & \vdots \\ a_{N1}\mathbf{B} & \cdots & a_{NM}\mathbf{B} \end{bmatrix}.$$

We define the $K$-th Kronecker power of $\mathbf{A}$ as $\mathbf{A}^{[K]} = \mathbf{A}^{[K-1]} \otimes \mathbf{A}$ and $\mathbf{A}^{[1]} = \mathbf{A}$.

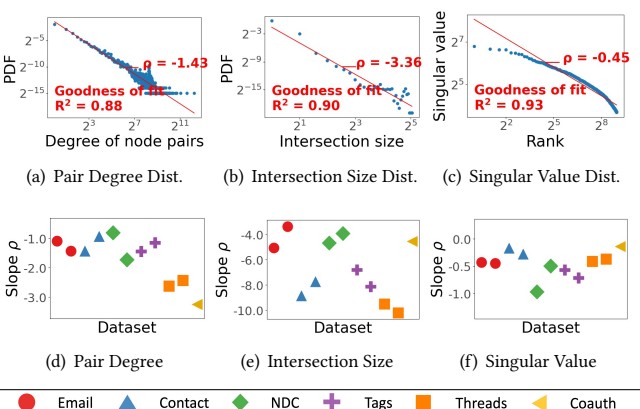

(a) Pair Degree Dist.

(b) Intersection Size Dist.

(c) Singular Value Dist.

(d) Pair Degree

(e) Intersection Size

(f) Singular Value

● Email ▲ Contact ◆ NDC ✚ Tags ■ Threads ◀ Coauth

**Figure 1: Discovery D1: Real-world Hypergraphs Follow Power-law Distributions. (a)-(c): The distributions of node pair degrees, intersection sizes, and singular values from email-Eu fit well with linear regressions on a log-log scale, indicated by $R^2$ scores close to 1. (d)-(f): The slopes tend to be similar within the same domain.**

## 3.3 Power-law and Log-logistic Distributions

**Power-law distributions** are frequently observed in various fields[15, 32]. Mathematically, a quantity $x$ follows a power-law distribution if its probability distribution function is of the form $p(x) \approx x^{-\alpha}$, where $\alpha$ is a constant. The **log-logistic distribution**, introduced in economics[7], occurs when the logarithm of a random variable ($\ln x$) follows a logistic distribution. A key characteristic of log-logistic distributions is that the odds ratios [12][1] derived from them follow power-law distributions, linking the two distributions.

## 3.4 Datasets

We consider eleven real-world hypergraphs from six different domains [3]: (a) **Email** [22, 30, 43] with nodes as email accounts and hyperedges as emails (i.e., the sender and receivers); (b) **Contact** [34, 41] with nodes as people and hyperedges as group interactions; (c) **Drugs (NDC)** with nodes as drug substances and classes and each hyperedge as the group contained in a drug; (d) **Tags** with nodes as tags and hyperedges as questions attached with relevant tags; (e) **Threads** with nodes as users and each hyperedge as the group discussing in a thread; (f) **Co-authorship** [40] with nodes as authors and each hyperedge as the coauthors of a publication. We use all hyperedges in each dataset, without filtering out duplicates or large-scale hyperedges. Their statistics are given in Table 2.

## 4 DISCOVERIES

In this section, we discover eight patterns across eleven real-world hypergraphs (described in Section 3.4), summarized in Figure 1,2 and Table 3. We first uncover (**D1**) power-law distributions in node pair degrees, intersection sizes, and singular values of incidence matrices, with a focus on distribution slopes. Then, we reveal that (**D2**) node degrees and hyperedge sizes exhibit log-logistic distributions. Lastly, we discover (**D3**) additional power-law patterns in clustering coefficients, density, and overlapness in egonets.

[1]The odds ratio is defined as $\mathrm{OddRatio}(x) := \frac{\mathrm{CDF}(x)}{1-\mathrm{CDF}(x)}$, where CDF is the cumulative distribution function.

## 4.1 D1: Power-law Distributions

Prior studies have revealed that node pair degrees [27], hyperedge intersection sizes [24], and singular values [24] exhibit heavy-tailed distributions. The **degree of a node pair** $i, j$ is defined as $d^{(2)}(i, j) := |e \in \mathcal{E} : i, j \subseteq e|$, i.e., the number of hyperedges containing that pair. **Intersection size** measures the overlap between hyperedges $e_i, e_j$ as $|e_i \cap e_j|$, while **singular values** are derived from the hypergraph's incidence matrix (see Section 3.1).[2]

We extend these findings to eleven real-world hypergraphs, including those with duplicated hyperedges. We first evaluate the fits to power-law or log-normal distributions[3] using log-likelihood ratios (LRs) to compare them against exponential distributions. Note that log-normal distributions are closely related to power-law distributions (see Section 3.3). Table 3 shows that the LRs are significantly greater than zero, indicating that power-law or log-normal distributions provide a better fit than exponential ones.

For these three distributions, we also evaluate the quality of linear regression fits ($R^2$ scores) on a log-log scale, commonly used to test power-law properties [12, 15]. Table 3 and Figure 1 show high $R^2$ scores near 1 (i.e., good linear regression fitting on a log-log scale), confirming power-law behavior. Similar slopes across hypergraphs within the same domain suggest domain-based similarities.

## 4.2 D2: Log-logistic Distributions

Recent studies reveal that both node degrees and hyperedge sizes exhibit heavy-tailed distributions rather than exponential ones [24]. However, they overlook the observed flatness at lower degrees or sizes, deviating from perfect power-law distributions. Our analysis of real-world hypergraphs suggests that, more precisely, both **node degrees** and **hyperedge sizes** follow *log-logistic* distributions.

Based on the relation between power-law and log-logistic distributions as discussed in Section 3.3, we investigate the odds ratios for node degrees and hyperedge sizes. Through linear regression on a log-log scale of these odds ratios, we find that the $R^2$ scores, indicative of fit quality, are close to 1. These power-law like distributions of odd ratios, in turn, imply that both node degrees and hyperedge sizes adhere to log-logistic distributions. Table 3 shows $R^2$ scores above 0.8 across all real-world hypergraphs (see Figure 2), with similar slopes within dataset domains.

## 4.3 D3: Additional Power-law Patterns

We present three new power-law patterns in egonets within real-world hypergraphs. An **egonet** for a central node $v$ is the set of hyperedges containing $v$, i.e., $\tilde{\mathcal{E}}_{\{v\}} := \{e \in \mathcal{E} : v \in e\}$.

**Clustering Coefficients.** We investigate the count of intersecting hyperedge pairs relative to the central node's degree in egonets. Since all hyperedges in an egonet already share the central node, we focus on pairs that also share additional nodes. Specifically, we compute $|\{\{e_i, e_j\} : \{e_i \cap e_j\} - \{v\} \neq \varnothing \land i \neq j \land e_i \in \tilde{\mathcal{E}}_{\{v\}} \land e_j \in \tilde{\mathcal{E}}_{\{v\}}\}|$. Since this is related to the clustering coefficient in bipartite graphs [38, 42, 44], reflecting 4-cycle statistics for nodes. we refer to this as the **clustering coefficient** (clustering coef.). As shown in

[2]Given the rapid decay of singular value distributions at the tail, we focus on top 50% singular values. For larger datasets, spec., tags, threads, and co-authorship, we use top 1,000, 1,000, and 500 singular values, respectively.

[3]We use a power-law distribution for node pair degrees and log-normal distributions for intersection sizes and singular values.

**Table 3:** Discoveries D1-D3: Evaluation of Power-law Fitness. For each dataset, we report the goodness of fit to power-law using the $R^2$ score of linear regression on a log-log scale, with the slope of the regression line. For the probability distributions regarding D1, we also compute log-likelihood ratios (LR) comparing fits to power-law or log-normal distributions against fits to exponential distributions. $R^2$ scores over 0.8, slopes with p-values under 0.05, and positive LRs are highlighted in bold.

| Dataset | D1. Power-law Dist. | | | | | | | | | D2. Log-logistic Dist. | | | | D3. Power-law Patterns | | | | | |
| | Pairdegree | | | Intersection | | | Singular Value | | | Degree | | Size | | Clustering Coef. | | Density | | Overlapness | |
| | $R^2$ | Slope | LR | $R^2$ | Slope | LR | $R^2$ | Slope | LR | $R^2$ | Slope | $R^2$ | Slope | $R^2$ | Slope | $R^2$ | Slope | $R^2$ | Slope |
|---|---|---|---|---|---|---|---|---|---|---|---|---|---|---|---|---|---|---|---|
| email-Enron | **0.8** | **-1.1** | **13** | **0.9** | **-5.1** | **22** | **1.0** | **-0.4** | **44** | **1.0** | **1.3** | **1.0** | **3.6** | **1.0** | **2.0** | 0.1 | **0.5** | 0.2 | **0.6** |
| email-Eu | **0.9** | **-1.4** | **1** | **0.9** | **-3.4** | **212** | **0.9** | **-0.4** | **158** | **1.0** | **0.7** | **0.9** | 2.0 | **0.9** | **1.8** | **0.8** | **1.2** | **0.8** | **1.3** |
| contact-primary | **0.9** | **-1.4** | **2** | **0.9** | **-8.8** | **8** | **0.9** | **-0.2** | **107** | **1.0** | **2.6** | **1.0** | **9.7** | **0.8** | **1.9** | **0.8** | **1.0** | **0.8** | **1.0** |
| contact-high | **0.8** | **-0.9** | **24** | **0.9** | **-7.7** | **20** | **0.9** | **-0.3** | **120** | **1.0** | **1.5** | **0.9** | **10.0** | **0.9** | **2.1** | **0.8** | **1.5** | **0.8** | **1.5** |
| NDC-classes | 0.6 | **-0.8** | **6** | **0.8** | **-4.7** | **6** | **0.9** | **-1.0** | **26** | **1.0** | **0.7** | **1.0** | **3.1** | **0.9** | **1.9** | 0.2 | **0.6** | 0.6 | **1.0** |
| NDC-substances | **0.8** | **-1.7** | **6** | **1.0** | **-3.9** | **40** | **1.0** | **-0.5** | **144** | **1.0** | **0.9** | **0.9** | **1.8** | 0.6 | **1.6** | 0.5 | 0.8 | **0.9** | **1.2** |
| tags-ubuntu | **0.8** | **-1.4** | **35** | **0.9** | **-6.8** | **17** | **1.0** | **-0.6** | **184** | **1.0** | **0.9** | **1.0** | **2.2** | **0.9** | **1.8** | **1.0** | **1.5** | **1.0** | **1.5** |
| tags-math | **0.8** | **-1.1** | **36** | **0.9** | **-8.1** | **5** | **1.0** | **-0.7** | **176** | **1.0** | **0.8** | **1.0** | **2.4** | **1.0** | **1.8** | **0.9** | **1.8** | **0.9** | **1.7** |
| threads-ubuntu | **0.9** | **-2.6** | **3** | **1.0** | **-9.5** | **3** | **1.0** | **-0.4** | **148** | **1.0** | **1.1** | **1.0** | **5.1** | **0.8** | **1.4** | **1.0** | **1.0** | **1.0** | **1.1** |
| threads-math | **0.9** | **-2.4** | **15** | **1.0** | **-10.2** | **16** | **1.0** | **-0.4** | **241** | **1.0** | **1.0** | **1.0** | **4.9** | **0.9** | **1.6** | **1.0** | **1.0** | **1.0** | **1.1** |
| coauth-geology | **1.0** | **-3.2** | **54** | **1.0** | **-4.5** | **13** | **1.0** | **-0.1** | **165** | **0.9** | **1.9** | **1.0** | **3.1** | **0.9** | **1.7** | **0.9** | **1.0** | **1.0** | 1.1 |

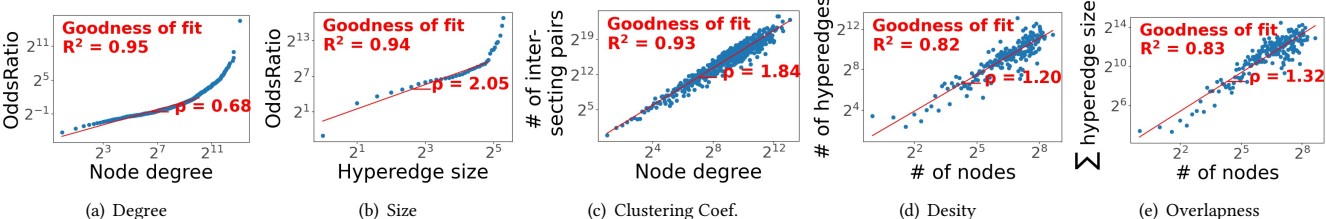

(a) Degree    (b) Size    (c) Clustering Coef.    (d) Desity    (e) Overlapness

**Figure 2:** Discoveries D2 and D3: Real-world Hypergraphs Follow Log-logistic Distributions and Exhibit Power-law Patterns. (a)-(b): The odds ratio functions of node degrees and hyperedge sizes are linear on a log-log scale, indicated by $R^2$ scores over 0.9. (c)-(e): The distributions of clustering coefficients, egonet density, and overlapness are also well-fitted by linear regression on a log-log scale. These distributions are from email-Eu.

Table 3 and Figure 2, the distribution of intersecting hyperedge pairs relative to the central node's degree follows a power-law pattern. It fits well with linear regression on a log-log scale with high $R^2$ scores close to 1.0 and slopes around 2.0, indicating a consistent intersection rate across egonets.

**Density and Overlapness of Egonets.** We further examine the relationship between the numbers of hyperedges and nodes within egonets. As illustrated in Table 3 and Figure 2, this relationship fits well with a linear regression model in a log-log scale, exhibiting a slope greater than 1. Given that the **density** [18, 27] of an egonet is defined as the ratio of the number of hyperedges to the number of nodes (i.e., $|\tilde{\mathcal{E}}_{\{v\}}|/|\bigcup_{e \in \tilde{\mathcal{E}}_{\{v\}}} e|$), our findings suggest the egonet density increases as the node count grows, with the rate of increase mirrored by the regression slope. We also examine the sum of hyperedge sizes in relation to the node count within egonets, which reveals a power-law pattern. The **overlapness** of egonets [27], defined as the ratio of the sum of hyperedge sizes to the number of nodes (i.e., $(\sum_{e \in \tilde{\mathcal{E}}_{\{v\}}} |e|)/|\bigcup_{e \in \tilde{\mathcal{E}}_{\{v\}}} e|$), also increases with more nodes, influenced by the slope. The slopes close to 1 indicate a slow rate of such increase in real-world hypergraphs.

## 5 PROPOSED HYPERGRAPH GENERATOR

In the previous section, we could observe multiple power-law patterns in real-world hypergraphs. Inspired by the Kronecker graph model [29], which is proven to produce multiple power-law patterns for conventional graphs, we introduce **HyRec**, a tractable and realistic hypergraph generative model.

### 5.1 Description of HyRec

We define the Kronecker product of two hypergraphs as the Kronecker product of their incidence matrices (see Section 3). For hypergraphs $\mathcal{G}$ and $\mathcal{H}$ with incidence matrices $I(\mathcal{G})$ and $I(\mathcal{H})$, their Kronecker product $\mathcal{G} \otimes \mathcal{H}$ is the hypergraph with incidence matrix $I(\mathcal{G}) \otimes I(\mathcal{H})$. Based on this, we present HyRec, a hypergraph model using the Kronecker product. Given an initiator hypergraph $\mathcal{G}$ and order $K$, $\text{HyRec}(\mathcal{G}, K) := \mathcal{G}^{[K]}$ is the hypergraph with $I(\mathcal{G})^{[K]}$, the $K$-th Kronecker power (see Section 3) of $I(\mathcal{G})$.

### 5.2 Theoretical Characteristics of HyRec

In this section, we derive several theoretical characteristics of HyRec, including multinomial distributions across various structural measures and evolutionary patterns that mirror those in real-world hypergraphs. This tractability is valuable, allowing for easier analysis and a better understanding of HyRec's behavior. It also facilitates parameter fitting (see Section 6.2).

**Preliminary: Multinomial Distributions.** Multinomial distributions are a generalization of binomial distributions. The parameters of a multinomial distribution are (1) $k$ for the number of event types, (2) $n$ for the number of (independent) trials, and (3) $p_i$ for the probability for the $i$-th event occurring at each trial, for each $i \in \{1, \cdots, k\}$, where $\sum_{i=1}^{k} p_i = 1$. After $n$ independent trials, the probability for the $i$-th event occurring exactly $c_i$ times for every $i \in \{1, \cdots, k\}$, where $c_1, \cdots, c_k \geq 0$ and $\sum_{i=1}^{k} c_i = n$, is $\frac{n!}{c_1! \cdots c_k!} p_1^{c_1} \cdots p_k^{c_k}$. It is a well-known fact that, with a careful choice of the parameters, multinomial distributions behave similarly to log-logistic and power-law distributions [5, 10, 29].

**Structural Patterns.** We prove that HyRec creates hypergraphs with several statistics following multinomial distributions. As mentioned above, multinomial distributions resemble log-logistic and power-law distributions, which are commonly observed in real-world hypergraphs (refer to Section 4).

**Theorem 1.** *HyRec($\mathcal{G}, K$) has multinomial distributions of (1) degrees, (2) hyperedge sizes, (3) pair degrees, and (4) intersection sizes.*
*Proof.* Refer to Appendix A.2 for the proof. ∎

**Theorem 2.** *In HyRec ($\mathcal{G}, K$), both singular values and singular vectors of its incidence matrix follow multinomial distributions.*
*Proof.* Refer to Appendix A.3 for the proof. ∎

**Evolutionary Patterns.** We prove the evolutionary patterns of HyRec, focusing on changes in density and (effective) diameter as the exponent $K$ of the Kronecker power increases, which can be considered as the hypergraph's growth over time.

**Theorem 3.** *In HyRec($\mathcal{G}, K$) where $I(\mathcal{G}) \in \{0, 1\}^{N_1 \times M_1}$ exhibits a $1 : \frac{\log M_1}{\log N_1}$ power-law relationship between the number of nodes and the number of hyperedges as $K$ increases.*
*Proof.* Refer to Appendix A.4 for the proof. ∎

**Theorem 4.** *If the initiator hypergraph $\mathcal{G}$ has a diameter $D$, the diameter of HyRec($\mathcal{G}, K$) is exactly $D$.*
*Proof.* Refer to Appendix A.5 for the proof. ∎

**Theorem 5.** *If the initiator hypergraph $\mathcal{G}$ has a diameter $D$, then the effective diameter of HyRec($\mathcal{G}, K$) converges to $D$ as $K$ increases.*
*Proof.* Refer to Appendix A.6 for the proof. ∎

### 5.3 Stochastic HyRec: Stochastic Version

We have so far applied the Kronecker power approach to a binary initiator matrix, which always produces the same hypergraph, limiting variability, a key property for (hyper)graph models in tasks like statistical testing [35]. To address this, we introduce a stochastic version of HyRec. Starting with an initiator matrix $\Theta \in [0, 1]^{N_1 \times M_1}$, we compute $\Theta^{[K]}$, where each $(i, j)$-th entry represents the probability of the $i$-th node being part of the $j$-th hyperedge. We then sample a hypergraph $\tilde{\mathcal{G}}$ by independently performing Bernoulli trials on each entry of $\Theta^{[K]}$, generating a binary incidence matrix $I(\tilde{\mathcal{G}})$ of $\tilde{\mathcal{G}}$. Appendix D provides examples showing how different probabilistic initiator matrices, including community and core-fringe structures, produce hypergraphs with diverse properties.

## 6 SINGFIT: FITTING TO REAL-WORLD HYPERGRAPHS

In the preceding analysis, we demonstrate that HyRec can replicate the diverse properties of real-world hypergraphs. *But how can we generate a Kronecker hypergraph that closely resembles a specific real-world hypergraph?* Specifically, how can we identify an initiator matrix (i.e., $\Theta$) that captures the underlying mechanisms? To answer this, we explore fitting the initiator matrix to the target hypergraph. Throughout this section, we focus on the stochastic version of HyRec, referred to simply as HyRec.

### 6.1 Challenges in Fitting HyRec

Fitting an initiator matrix poses the following challenges:

**C1. Computational Cost of Alignment.** Identifying correspondences between nodes or hyperedges of input and generated hypergraphs requires considering all possible ($|\mathcal{V}|! \times |\mathcal{E}|!$) permutations of nodes and hyperedges. Thus, directly aligning incidence matrices faces computational challenges.

**C2. Non-Differentiability of Generation.** The stochastic version of HyRec uses probability matrices (i,e., $\Theta^{[K]}$) to generate hypergraphs, independently drawing each entry to form binary matrices (i.e., $I(\tilde{\mathcal{G}})$). This process causes a discrepancy between the characteristics (e.g., singular values) of probability matrices and those of binarized matrices. Additionally, the non-differentiable nature of the sampling process poses challenges for parameter fitting.

**C3. Density of Probability Matrix.** Naively sampling a hypergraph $\tilde{\mathcal{G}}$ from the probability matrix $\Theta^{[K]}$ leads to high computational and memory overhead, as Bernoulli sampling must be applied to every possible connection between nodes and hyperedges, resulting in a complexity of $O(|\mathcal{V}||\mathcal{E}|)$.

### 6.2 Strategies for Overcoming Fitting Challenges with SingFit

We propose SingFit, an initiator-matrix fitting algorithm that addresses the above challenges with three solutions.

**S1: Singular-Value Matching.** To avoid high alignment costs (Challenge **C1**), we aim to match statistics that do not require aligning nodes and hyperedges between input and generated hypergraphs. One of the statistics is the incidence-matrix singular values of the input hypergraph **G** and the generated hypergraph $\tilde{\mathbf{G}}$ (singular values are invariant to the row and column orders), quantified by the following loss function:

$$\mathcal{L}_\sigma = \sum_{i=1}^{|\sigma(\mathbf{G})|} \left(\sigma(\mathbf{G})_i - \sigma(\tilde{\mathbf{G}})_i\right)^2 / |\sigma(\mathbf{G})| \tag{1}$$

Here, $\sigma(\cdot)$ is a function that computes the singular values of the incident matrix of a given hypergraph, sorted in descending order. The symbol $|\sigma(\mathbf{G})|$ represents the number of singular values, i.e., the rank of the incident matrix of **G**.

To further improve the model's ability, we include additional loss terms $\mathcal{L}_d$ and $\mathcal{L}_s$ to learn the distributions of node degrees and hyperedge sizes. Here, $d(\cdot)$ and $s(\cdot)$ compute the node degrees and hyperedge sizes, respectively, sorted in descending order. Since sorted, they are independent of node and hyperedge alignments. As a result, we fit the initiator matrix by minimizing the combined loss function $\mathcal{L} = \mathcal{L}_\sigma + \lambda_d \mathcal{L}_d + \lambda_s \mathcal{L}_s$, where $\lambda_d$ and $\lambda_s$ are positive weights controlling the influence of $\mathcal{L}_d$ and $\mathcal{L}_s$ respectively. Our preliminary experiments reveal that these additional terms are especially effective for hypergraphs with small hyperedges, including those from the contact or tag domains.

**S2: Differentiable Sampling with Gumbel-Softmax.** To address the discrepancy between the probability matrix $\Theta^{[K]}$ and the binary incidence matrix $I(\tilde{\mathcal{G}})$, which is generated after the fitting process (Challenge **C2**), we propose the use of Gumbel-Softmax [20]. It bridges the gap by simulating the generation process during parameter fitting, ensuring the model's output aligns more closely with the target binary-valued structure.

Formally, we derive a differentiable binary matrix $\hat{X}$ from a probability matrix $X$ of the same size as follows:

$$X'^{(c)}_{i,j} = \frac{\exp\left(\frac{\log(X_{i,j})+g^{(c)}_{i,j}}{\tau}\right)}{\exp\left(\frac{\log(1-X_{i,j})+g^{(0)}_{i,j}}{\tau}\right) + \exp\left(\frac{\log(X_{i,j})+g^{(1)}_{i,j}}{\tau}\right)},$$

$$\hat{X}_{i,j} = \mathbf{argmax}_{c\in\{0,1\}}\left(X'^{(c)}_{i,j}\right) + X'^{(1)}_{i,j} - \mathbf{sg}\left(X'^{(1)}_{i,j}\right).$$

Here, $X'^{(c)}_{i,j}$ represents the probability of the sampled $(i,j)$-th element being $c \in \{0,1\}$, and $g^{(c)}_{i,j}$ is a random sample drawn independently from the Gumbel$(0,1)$ distribution for each entry $(i,j)$ and $c \in \{0,1\}$. The softmax temperature $\tau$ controls the closeness of entries to binary values. The stop gradient operator $\mathbf{sg}$ ensures $\hat{X}_{i,j}$ takes binary values, while allowing gradient calculation as $\nabla_X \hat{X}_{i,j} = \nabla_X X'^{(1)}_{i,j}$. Thus this solves Challenge **C2**.

**S3: Acceleration Using Kronecker Product Properties.** To address high computational costs of handling the probability matrix $\Theta^{[K]}$ (Challenge **C3**), we leverage Kronecker product properties of distributions of singular values, node degrees and hyperedge sizes. In the proofs of Theorem 1 (see Appendix A.2) and Theorem 2 (see Appendix A.3), we show that these properties of the Kronecker hypergraph can be expressed as kronecker powers of the corresponding properties of the initiator hypergraph. This allows us to approximate these properties sampled from $\Theta^{[K]}$ using unit sampling, where we decompose the Kronecker power to the $K$ into $L$ units with smaller exponents and compute the properties sampled from these smaller units.

Formally, when $I(\mathcal{G})$ can be decomposed using SVD as $I(\mathcal{G}) = U_1\Sigma_1V_1^\top$, the singular values $\Sigma_K$ of $I(\mathcal{G})^{[K]}$, are expressed as:

$$\Sigma_K = \underbrace{(\Sigma_1 \otimes \Sigma_1 \otimes \cdots \otimes \Sigma_1)}_{K \text{ times}} = \Sigma_1^{[K]} \quad (\because Appendix\ A.3)$$

$$= \underbrace{(\Sigma_1 \otimes \cdots \otimes \Sigma_1)}_{S_1 \text{ times}} \otimes \underbrace{(\Sigma_1 \otimes \cdots \otimes \Sigma_1)}_{S_2 \text{ times}} \otimes \cdots \otimes \underbrace{(\Sigma_1 \otimes \cdots \otimes \Sigma_1)}_{S_L \text{ times}}$$

$$= \Sigma_1^{[S_1]} \otimes \Sigma_1^{[S_2]} \otimes \cdots \otimes \Sigma_1^{[S_L]} \quad (\text{where } \sum_{i=1}^{L} S_i = K)$$

$$= \Sigma_{S_1} \otimes \Sigma_{S_2} \otimes \cdots \otimes \Sigma_{S_L},$$

Here, $L$ is the number of units each with size $S_i$. Thus, we approximate the singular values $\Sigma_K$ sampled from $\Theta^{[K]}$ by calculating the singular values $\Sigma_{S_i}$ sampled from smaller $\Theta^{[S_i]}$ and then compute their kronecker products. This substantially reduces the computational complexity from $O(\min(N_1^2\ M_1, N_1M_1^2)^K)$ to $O(L \cdot \min(N_1^2M_1, N_1M_1^2)^{K/L})$ (refer to Section 7.4 for empirical results and Appendix B for complexity analysis). The same approach can be applied to compute node degrees and hyperedge sizes; see our finding in the proof of Theorem 1 (Appendix A.2).

## 6.3 Description of SINGFIT

Algorithms 1 and 2 outline the fitting and generation procedures, respectively. The process involves sampling (i.e., binarizing) unit matrices across $L$ iterations, where $L = 1$ represents the naive approach using the full probabilistic matrix. When fitting the initiator, we employ the Gumbel-Softmax technique for sampling, which enables gradient calculation. Singular values are computed from these sampled matrices and combined using Kronecker products

---

**Algorithm 1:** Initiator Fitting of HyRec: SINGFIT

**Input** : (1) Incidence matrix of hypergraph $G \in \mathbb{R}^{N \times M}$,
 (2) Size of the initiator matrix $N_1$ and $M_1$,
 (3) Number of units $L$,
 (4) Number of iterations $E$
**Output**: Initiator $\Theta \in \mathbb{R}^{N_1 \times M_1}$

1 $K = \lceil MAX(\log_{N_1} N, \log_{M_1} M) \rceil$
2 $S \leftarrow$ GenerateUnitSizes$(K, L)$        ▷ See below
3 $\tilde{\sigma} \leftarrow I^{1 \times 1}$
4 $\Theta \leftarrow softplus(\Theta_{init})$ where $\Theta_{init} \sim \mathcal{U}(\mathbb{R})$
5 **for each** epoch $e = 1, \cdots, E$ **do**
6   **for each** unit $i = 1, \cdots, L$ **do**
7     $\hat{\Theta}^{[S_i]} \leftarrow$ **GumBel**$(\Theta^{[S_i]})$
8     $\tilde{\sigma} \leftarrow \tilde{\sigma} \otimes \sigma(\hat{\Theta}^{[S_i]})$    ▷ $\sigma(\cdot)$ computes singular values
9     $\tilde{d} \leftarrow \tilde{d} \otimes d(\hat{\Theta}^{[S_i]})$    ▷ $d(\cdot)$ computes node degrees
10     $\tilde{s} \leftarrow \tilde{s} \otimes s(\hat{\Theta}^{[S_i]})$    ▷ $s(\cdot)$ computes hyperedge sizes
11   Update $\Theta$ by $\nabla_\Theta \mathcal{L}(= \mathcal{L}_\sigma + \lambda_d \mathcal{L}_d + \lambda_s \mathcal{L}_s)$    ▷ Eq. (1)
12 **return** $\Theta$

1 GenerateUnitSizes$(K, L)$
2   $S \leftarrow$ List of length $L$
3   **for each** unit $i = 1, \cdots, L$ **do**
4     $S_i \leftarrow \lfloor \frac{K}{L} \rfloor$
5     **if** $i \leq mod(K, L)$ **then** $S_i \leftarrow S_i + 1$ ;
6   **return** S      ▷ Ensures that $\sum_{i=1}^{L} S_i = K$

---

**Algorithm 2:** Hypergraph Generation in HyRec

**Input** : (1) Initiator matrix $\Theta \in \mathbb{R}^{N_1 \times M_1}$,
 (2) order $K$,
 (3) Number of units $L$
**Output**: A generated hypergraph $\tilde{G}$

1 $S \leftarrow$ GenerateUnitSizes$(K, L)$      ▷ Algorithm 1
2 **for each** unit $i = 1, \cdots, L$ **do**
3   $\hat{\Theta}^{[S_i]} \leftarrow$ Sample i.i.d. from $\Theta^{[S_i]}$
4   $\tilde{G} \leftarrow \tilde{G} \otimes \hat{\Theta}^{[S_i]}$
5 **return** $\tilde{G}$

---

to produce the complete singular value set. Node degrees and hyperedge sizes are computed similarly. For the generation phase, when the sampled unit incidence matrices are sparse, we focus on '1' entries to improve computational and memory efficiency. Efficiency-related experimental results are presented in Section 7.4 and Appendix G, while a detailed complexity analysis can be found in Appendix B (Table 7 and 8). In all experiments, we use at least two units (i.e., $L \geq 2$) and maintain the same number of units ($L$) for both training and generation.

## 7 EXPERIMENTAL RESULTS

In this section, we review our experiments, whose results demonstrate the effectiveness of HyRec.

## 7.1 Experimental Settings

**Competitors.** We consider 5 baseline generators: **HyperCL** [27], **HyperFF** [23], **HyperPA** [13], **HyperLap** [27], and **THera** [21],

**Table 4: HYREC Fits Real-World Hypergraphs**

(a) **HYREC obeys the patterns in Section 4**: HYREC (green) closely resembles the properties of real-world hypergraphs (black).

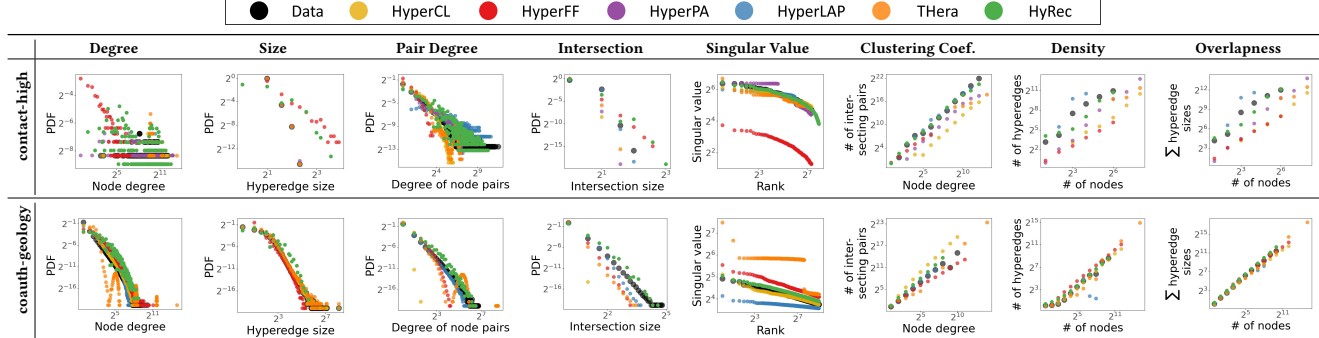

(b) **HYREC is accurate and parsimonious**: Across eleven datasets, HYREC ranks within the top three for most properties and ranks second on average. It has the second-fewest input parameters, following HyperFF, which ranks last. **Note that the top-performing model requires three orders of magnitude more parameters than HYREC.** The best, second-best, and third-best performance are highlighted in blue, green, and yellow, respectively.

| | # Input Parameters | | Average Ranking (Across 11 Hypergraph Datasets) | | | | | | | | | |
|---|---|---|---|---|---|---|---|---|---|---|---|---|
| | Min | Max | Degree | Size | Pair Deg. | Intersect. | Singular Value | Clustering Coef. | Density | Overlapness | Effective Diam. | Average |
| HyperCL | 11,028 | 2,852,295 | **1.455** | **1.000** | 3.455 | 4.909 | *2.364* | 4.545 | 3.636 | 4.364 | *3.455* | 3.242 |
| HyperFF | **2** | **2** | 4.909 | 4.818 | 5.000 | 4.273 | 5.182 | 4.182 | 4.273 | 3.909 | 3.818 | 4.485 |
| HyperPA | 11,028 | 3.727 | *3.000* | 3.727 | *3.000* | 4.273 | 4.545 | 4.636 | 4.000 | 4.273 | *3.545* | 3.889 |
| HyperLAP | 11,028 | 2,852,295 | *2.636* | **1.000** | 2.636 | **1.636** | 3.182 | **1.364** | *3.273* | *2.727* | 4.727 | **2.576** |
| THera | 10,889 | 1,591,170 | 4.909 | 4.091 | 3.091 | *2.364* | 4.545 | *2.636* | *3.273* | 3.182 | 3.909 | 3.556 |
| **HYREC** | *15* | *882* | 4.091 | 4.818 | 3.818 | *3.545* | **1.182** | 3.636 | **2.545** | **2.545** | **1.545** | *3.081* |

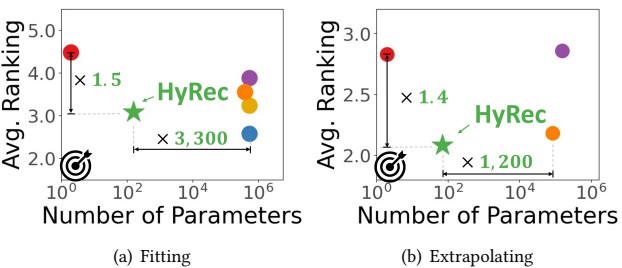

(a) Fitting  (b) Extrapolating

**Figure 3: HYREC Demonstrates Superior Performance with Fewer Input Parameters: HYREC performs well in fitting and extrapolating real-world hypergraphs with a relatively small number of input parameters (in terms of the number of scalars), proving its efficiency and effectiveness. The rankings are averaged over all 9 patterns and 11 real-world hypergraphs, with lower values indicating better performance.**

with their features summarized in Table 1. HyperCL and HyperLAP rely on node degree and hyperedge size distributions as inputs. HyperPA is limited to hyperedges of size under 20 and requires distributions for both hyperedge sizes and the number of new hyperedges for nodes. THera also demands hyperedge size distributions. HyperFF is based on the forest fire model controlled by two parameters. While more generators [2, 8] could be considered for comparison, we focus on competitive open-source generators that have been shown to reproduce realistic structural properties. Detailed hyperparameter search spaces are provided in Appendix C.

**Evaluation.** We evaluated HYREC's performance in replicating **9 real-world patterns** (spec., those discussed in Section 4 and the

effective diameters of hypergraphs [24]). We consider the **11 real-world hypergraphs** from six domains (email, contact, drug, tags, threads, and co-authorship) described in Section 3.4. We evaluate the goodness of fit using the Kolmogorov-Smirnov ($D$-statistic) for the probability density distributions of degree, size, pair degree, and intersection size; Root Mean Square Error (RMSE) for singular values, clustering coefficients, density, and overlapness[4]; and relative difference for the effective diameter which is a scalar value. Hypergraphs are generated once per model, per parameter set in the search space, and per dataset. We rank each generator's fit for each pattern and average these rankings across the 11 datasets.

### 7.2 Fitting to Real-World Hypergraphs

We visually and statistically test how accurately hypergraphs generated by HYREC align with the distribution patterns of real-world hypergraphs. Table 4(a) visually confirms that HYREC produces distributions closely resembling real-world hypergraphs. Table 4(b) reports the average rankings of generators in matching each of the nine properties across eleven real-world datasets. HYREC ranks within the top three for six properties, demonstrating strong alignment with most properties. Although HYREC underperforms some baselines in terms of node degrees and hyperedge sizes, it is important to note that these baselines (except for HyperFF) directly rely on detailed node-degree and hyperedge-size distributions as inputs, making their strong performance on these properties unsurprising. This adds complexity, as they require detailed statistics for every generation. In contrast, HYREC uses only an initiator matrix fitted

---

[4]When computing RMSE of $y$ values (e.g., singular values or intersecting pairs in egonets), we consider only the intersection of the $x$ values (e.g., ranks or central node's degree in egonets) from the generator outputs and the ground-truth dataset. For clustering coefficients, density, and overlapness, this process is applied after a logarithmic binning of $x$ values.

## Table 5: HᴙRᴇᴄ Extrapolates Real-World Hypergraphs

(a) **HᴙRᴇᴄ predicts hypergraph growth accurately:** After fitting to a past snapshot of the NDC-substances dataset, HᴙRᴇᴄ (green) accurately predicts the properties of its future snapshot (which is the entire dataset).

(b) **HᴙRᴇᴄ is accurate in extrapolation, even with a small number of input parameters**: HᴙRᴇᴄ performs **overall best** in extrapolation, and notably it requires two orders of magnitude fewer parameters than the second-best model (THera). Note that HyperCL and HyperLAP are inapplicable to extrapolation. The best, second-best, and third-best performance are highlighted in blue, green, and yellow, respectively.

| | # Input Parameters | | | Average Ranking (Across 11 Hypergraph Datasets) | | | | | | | | | |
|---|---|---|---|---|---|---|---|---|---|---|---|---|---|
| | **Min** | **Max** | | **Degree** | **Size** | **Pair Deg.** | **Intersect.** | **Singular Value** | **Clustering Coef.** | **Density** | **Overlapness** | **Effective Diam.** | **Average** |
| HyperFF | 2 | 2 | | 2.818 | 2.909 | 3.273 | 2.818 | 2.909 | 1.818 | 2.909 | 2.909 | 3.091 | 2.828 |
| HyperPA | 466 | 1,167,675 | | 2.636 | 1.818 | 3.273 | 3.182 | 3.636 | 2.818 | 3.000 | 3.091 | 2.273 | 2.859 |
| THera | 349 | 537,114 | | 3.000 | 2.273 | 1.636 | 1.909 | 1.636 | 2.909 | 2.000 | 2.182 | 2.091 | 2.182 |
| **HᴙRᴇᴄ** | 12 | 396 | | 1.545 | 2.545 | 1.818 | 2.091 | 1.818 | 2.455 | 2.091 | 1.818 | 2.545 | 2.081 |

### Table 6: HᴙRᴇᴄ's Efficiency Gains with Increased Units.

| Number of Units (L) | Avg. Unit Matrix Size ($|\Theta^{[S_i]}|$) | Fitting Time (ms) | Generation Time (ms) |
|---|---|---|---|
| L = 1 | 85,766,121 | 378.983 | 157,656.327 |
| L = 2 | 9,261 | 23.697 | 19,619.597 |
| L = 3 | 441 | 28.730 | 5,369.581 |
| L = 4 | 231 | 42.813 | 10,151.488 |
| L = 5 | 105 | 48.959 | 7,815.624 |

by SɪɴɢFɪᴛ and a few scalars while offering both tractability and extrapolation ability, as shown in Table 1. HyperPA is inapplicable when the node count is too small (email-Enron) and runs out of memory when the hypergraph size is too large (coauth-geology), resulting in it ranking last in these cases. Notably, as illustrated in Figure 3, HᴙRᴇᴄ provides the best balance between performance and input parameter size (in terms of the number of scalars). We also analyze how HᴙRᴇᴄ's performance is affected by hyperparameters (e.g., the initiator matrix size and the unit number) in Appendix E.

### 7.3 Extrapolating Real-World Hypergraphs

We evaluate HᴙRᴇᴄ's extrapolation ability of HᴙRᴇᴄ in forecasting the evolution of hypergraph properties. We fit HᴙRᴇᴄ using the hyperedges up until the first 50% of the nodes appear, and we test it against the full original hypergraph. The same protocol is applied to HyperPA, HyperFF, and THera, but by their design, HyperLAP and HyperCL cannot predict beyond the input data. Table 5(a) visually confirms that HᴙRᴇᴄ is able to closely mirror a given hypergraph (past) and accurately predict its future. Table 5(b) and Figure 3 present that HᴙRᴇᴄ ranks first on average while requiring two orders of magnitude fewer input parameters (in terms of the number of scalars) than the second-best one. HᴙRᴇᴄ also achieves the best performance for extrapolation based on a snapshot with the first 25% of nodes, as shown in Appendix F.

### 7.4 Efficiency in Fitting Large Hypergraphs

Unit sampling, described in Section 6.2, significantly improves the efficiency of SɪɴɢFɪᴛ. Table 6 shows that increasing the number of units (i.e., $L$) in the contact-primary dataset reduces the total size of unit matrices and the runtime for both fitting and generation.[5] In this experiment, the initiator matrix is fixed at 3×7, , with a Kronecker power of order 6. Additionally, we compare the runtime of all models for generation in Appendix G, which demonstrates the efficiency of HᴙRᴇᴄ.

## 8 CONCLUSIONS

In this study, we uncover eight power-law-related patterns in real-world hypergraphs and introduce HᴙRᴇᴄ, a generative model leveraging the Kronecker product to replicate these patterns. We mathematically demonstrate that HᴙRᴇᴄ captures both structural and evolutionary patterns. Additionally, we develop SɪɴɢFɪᴛ, a fast and space-efficient algorithm for fitting HᴙRᴇᴄ to given hypergraphs. Our experiments on eleven real-world hypergraphs confirm the model's efficacy in fitting and forecasting hypergraph properties. We also discuss limitations and propose future research directions in Appendix H. Our contributions are summarized as follows:

- **Discoveries**: Identification of eight power-law-related patterns in real-world hypergraphs (Figures 1-2 and Table 3).
- **Model**: Design of HᴙRᴇᴄ, a tractable and realistic (Figure 3 and Tables 4-5) generative model supported by SɪɴɢFɪᴛ.
- **Proofs**: Mathematical validation that HᴙRᴇᴄ adheres to these identified patterns (Theorems 1 and 2).

For reproducibility, our code and data are available at [1].

---

[5]Since units are processed serially (while operations within each unit are parallelized), the total time may increase slightly with more units.

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

# APPENDIX

# A THEORETICAL PROOFS OF HYREC

In this section, we provide the mathematical proofs for the theorems introduced in Section 5, establishing the theoretical characteristics of HyRec.

## A.1 Notations

We begin with introducing several notations.

- $v_i$: node in $\mathcal{G}$ for the $i$-th row of its incidence matrix $I(\mathcal{G})$.
- $e_i$: hyperedge for the $i$-th column of $I(\mathcal{G})$.
- $v_{i,j}$: node in $\mathcal{G} \otimes \mathcal{H}$ for the $((i-1)N_2 + j)$-th row of its incidence matrix $I(\mathcal{G} \otimes \mathcal{H})$ where $I(\mathcal{G}) \in \{0,1\}^{N_1 \times M_1}$ and $I(\mathcal{H}) \in \{0,1\}^{N_2 \times M_2}$.
- $e_{i,j}$: hyperedge for the $((i-1)M_2 + j)$-th column of $I(\mathcal{G} \otimes \mathcal{H})$.
- $v_{i_1,\cdots,i_K}$: node in HyRec$(\mathcal{G}, K)$ for the $\left(\sum_{k=1}^{K} \left((i_k-1)\cdot N^{K-k}\right)+1\right)$-th row of its incidence matrix $I(\mathcal{G})^{[K]}$ where $I(\mathcal{G}) \in \{0,1\}^{N \times M}$.
- $e_{i_1,\cdots,i_K}$: hyperedge for the $\left(\sum_{k=1}^{K} \left((i_k-1)\cdot M^{K-k}\right)+1\right)$-th column of $I(\mathcal{G})^{[K]}$.

## A.2 Proof of Theorem 1

**Theorem.** HyRec$(\mathcal{G}, K)$ has multinomial distributions of (1) degrees, (2) hyperedge sizes, (3) pair degrees, and (4) intersection sizes.

**Proof.** By Theorem 5 in [29], we can easily show that (1) degrees and (2) hyperedge sizes follow multinomial distributions. Regarding (3) pair degrees, let $S_1$ denote the multiset[6] $\{\omega_{i,i'} : i, i' \in \{1,\cdots,N_1\} \wedge i \neq i'\}$ of node pair degrees in $\mathcal{G}_1$ and $S_2$ denote the multiset $\{\bar{\omega}_{j,j'} : j, j' \in \{1,\cdots,N_2\} \wedge j \neq j'\}$ of node pair degrees in $\mathcal{G}_2$, where $I(\mathcal{G}_1) \in \{0,1\}^{N_1 \times M_1}$ and $I(\mathcal{G}_2) \in \{0,1\}^{N_2 \times M_2}$. If a pair of nodes $v_i$ and $v'_i$ in $\mathcal{G}_1$ has pair degree $\omega_{i,i'}$, and a pair of nodes $v_j$ and $v'_j$ in $\mathcal{G}_2$ has pair degree $\bar{\omega}_{j,j'}$, then a pair of nodes $v_{i,j}$ and $v_{i',j'}$ in $\mathcal{G}_1 \otimes \mathcal{G}_2$ has pair degree $\omega_{i,i'}\bar{\omega}_{j,j'}$. Thus, after the $K-1$ kronecker product operations, HyperK$(\mathcal{G}_1, K)$ has the multiset of node pair degrees $\{\omega_{i_1,j_1} \cdots \omega_{i_K,j_K} : i_1,\cdots,i_K, j_1,\cdots,j_K \in \{1,\cdots,N_1\} \wedge (i_1,\cdots,i_K) \neq (j_1,\cdots,j_K)\}$. Let $s_1,\cdots,s_l$ be the distinct elements in $S_1$, and $o_k$ be the number of occurrences of $s_k$ in $S_1$. Then, the multiset of node pair degrees in HyperK$(\mathcal{G}_1, K)$ follows multinomial distribution where each node pair degree $s_1^{c_1} \cdots s_l^{c_l}$ (where $c_1,\cdots,c_l$ are non-negative integers and $\sum_{i=1}^{l} c_i = K$) occurs with a probability proportional to $\frac{K!}{c_1!\cdots c_l!} o_1^{c_1} \cdots o_l^{c_l}$. Since (4) intersection sizes of a pair of hyperedge $j$ and $j'$ of $\mathcal{G}_1$ is the number of common entries between $j$-th column and $j'$-th column of $\mathcal{G}_1$, the above proof is applied similarly. ∎

## A.3 Proof of Theorem 2

**Theorem.** In HyRec $(\mathcal{G}, K)$, both singular values and singular vectors of its incidence matrix follow multinomial distributions

**Proof.** Let the singular value decomposition (SVD) of the incidence matrix $I(\mathcal{G}) \in \{0,1\}^{N_1 \times M_1}$ of the initiator hypergraph $\mathcal{G}$ be $\mathbf{U}\Sigma\mathbf{V}^\top$, where $\mathbf{U} \in \mathbb{R}^{N_1 \times R}$, $\Sigma \in \mathbb{R}^{R \times R}$, and $\mathbf{V} \in \mathbb{R}^{M_1 \times R}$. By the properties of Kronecker product [25], the SVD of $I(\mathcal{G})^{[K]}$ is

$\mathbf{U}^{[K]}\Sigma^{[K]}(\mathbf{V}^{[K]})^\top$. Thus, HyRec$(\mathcal{G}, K)$ has multinomial distributions of both singular values and singular vectors. ∎

[6]A *multiset* generalizes a *set* by allowing duplicate elements.

## A.4 Proof of Theorem 3

**Theorem.** In HyRec$(\mathcal{G}, K)$ where $I(\mathcal{G}) \in \{0,1\}^{N_1 \times M_1}$ exhibits a $1 : \frac{\log M_1}{\log N_1}$ power-law relationship between the number of nodes and the number of hyperedges as $K$ increases.

**Proof.** The node count $N(K)$ and the hyperedge count $E(K)$ in HyRec$(\mathcal{G}, K)$ become $N_1^K$ and $M_1^K$, respectively. Consequently, the relationship between $E(K)$ and $N(K)$ satisfies $E(K) = M_1^K = (N_1^K)^a = N(K)^a$ where $a = \frac{\log M_1}{\log N_1}$. ∎

## A.5 Proof of Theorem 4

We begin with presenting Lemma 1 and Lemma 2, which are used for proofs.

**Lemma 1.** The following claims hold:
- There exists a hyperedge containing $v_{i,j}$ and $v_{k,l}$ in $\mathcal{G} \otimes \mathcal{G}'$ if and only if (a) there is a hyperedge containing both $v_i$ and $v_k$ in $\mathcal{G}$, and (b) there is a hyperedge containing both $v_j$ and $v_l$ in $\mathcal{G}'$.
- The hyperedge $e_{p,q}$ in $\mathcal{G} \otimes \mathcal{G}'$ contains $v_{i,j}$ if and only if (a) $e_p$ in $\mathcal{G}$ contains $v_i$, and (b) $e'_q$ in $\mathcal{G}'$ contains $v'_j$.

**Proof.** The claims are straightforwardly deduced from the definition of the Kronecker product. ∎

**Lemma 2.** If two hypergraphs $\mathcal{G}$ and $\mathcal{G}'$ each have a diameter at most $D$, then $\mathcal{G} \otimes \mathcal{G}'$ also has a diameter at most $D$.

**Proof.** Consider two arbitrary nodes $v_i, v_k$ in $\mathcal{G}$ and two arbitrary nodes $v'_j, v'_l$ in $\mathcal{G}'$. Let $a$ be the distance between node $v_i$ and $v_k$ in $\mathcal{G}$, and $b$ be the distance between node $v'_j$ and $v'_l$ in $\mathcal{G}'$. Then, there is a path $(e_{p_1},\cdots,e_{p_a})$ between $v_i$ and $v_k$ in $\mathcal{G}$, and there is a path $(e'_{q_1},\cdots,e'_{q_b})$ between $v'_j$ and $v'_l$ in $\mathcal{G}'$. By Lemma 1, in $\mathcal{G} \otimes \mathcal{G}'$, the node $v_{i,k}$ is contained in the hyperedge $e_{p_1,q_1}$, and the node $v_{j,l}$ is contained in the hyperedge $e_{p_a,q_b}$. Additionally, for every $k \leq \max(a,b)-1$, $e_{p_{\min(k,a)},q_{\min(k,b)}} \cap e_{p_{\min(k+1,a)},q_{\min(k+1,b)}} \neq \emptyset$ holds. Therefore, a path $(e_{p_{\min(1,a)},q_{\min(1,b)}},\cdots,e_{p_{\min(\max(a,b),a)},q_{\min(\max(a,b),b)}})$ exists between $v_{i,k}$ and $v_{j,l}$ in $\mathcal{G} \otimes \mathcal{G}'$. Since $\max(a,b) \leq D$, the distance between any two nodes in $\mathcal{G} \otimes \mathcal{G}'$ is at most $D$. ∎

**Theorem.** If the initiator hypergraph $\mathcal{G}$ has a diameter $D$, the diameter of HyRec$(\mathcal{G}, K)$ is exactly $D$.

**Proof.** From Lemma 2, we can easily show the diameter of HyRec$(\mathcal{G}, K)$ is at most $D$ by employing induction on $K$. Since the diameter of $\mathcal{G}$ is $D$, there exists a pair of nodes $(v_i, v_j)$ such that the distance between $v_i$ and $v_j$ is exactly $D$. Suppose the diameter of HyRec$(\mathcal{G}, K)$ is at most $D-1$. Then, there is a path $(e_{p_{11},\cdots,p_{1K}}, e_{p_{21},\cdots,p_{2K}}, \cdots, e_{p_{I1},\cdots,p_{IK}})$ between $v_{i,\cdots,i}$ and $v_{j,\cdots,j}$ in HyRec$(\mathcal{G}, K)$, whose length $l$ is at most $D-1$. Since HyRec$(\mathcal{G}, K)$ is equivalent to HyRec$(\mathcal{G}, K-1) \otimes \mathcal{G}$, $(e_{p_{1K}}, e_{p_{2K}}, \cdots, e_{p_{IK}})$ is a valid path between $v_i$ and $v_j$ in $\mathcal{G}$, and its length is at most $D-1$, which is a contradiction. ∎

## A.6 Proof of Theorem 5

**Theorem**. *If the initiator hypergraph $\mathcal{G}$ has a diameter $D$, then the effective diameter of $\textsc{HyRec}(\mathcal{G}, K)$ converges to $D$ as $K$ increases.*

**Proof.** Suppose we randomly select a node $a = (v_{a_1, \cdots, a_K})$ and a node $b = (v_{b_1, \cdots, b_K})$ from $\textsc{HyRec}(\mathcal{G}, K)$, where $I(\mathcal{G}) \in \{0, 1\}^{N_1 \times M_1}$. Since the diameter of $\mathcal{G}$ is $D$, there exists a pair of nodes $(v_i, v_j)$ such that the distance between $v_i$ and $v_j$ is exactly $D$, then with probability $1 - (1 - \frac{1}{N_1^2})^K$, there is some index $k \in \{1, \cdots, K\}$ s.t $(a_k, b_k) = (i, j)$. Thus, the probability that the distance between two nodes in $\textsc{HyRec}(\mathcal{G}, K)$ is exactly $D$ is at least $1 - (1 - \frac{1}{N_1^2})^K$. Since there exists $K_0 = 3N_1^2 \in \mathbb{N}_k$ such that $1 - (1 - \frac{1}{N_1^2})^K > 1 - \frac{1}{e^3} > 0.9$ for all $K > K_0$, the effective diameter of $\textsc{HyRec}(\mathcal{G}, K)$ converges to $D$ as $K$ increases. ∎

**Table 7: Generation Time and Memory Complexity Comparison**

| Generator | Time Complexity | Memory Complexity |
|---|---|---|
| HyperCL | $O(log_2|\mathcal{V}| \cdot \sum_{e \in \mathcal{E}} |e|)$ | $O(|\mathcal{V}| + \sum_{e \in \mathcal{E}} |e|)$ |
| HyperFF | $O(|\mathcal{V}| \cdot \sum_{e \in \mathcal{E}} |e|)$ | $O(|\mathcal{V}| + \sum_{e \in \mathcal{E}} |e|)$ |
| HyperPA | $O(\sum_{e \in \mathcal{E}} log_2 \binom{|\mathcal{V}|}{|e|})$ | $O(\sum_{e \in \mathcal{E}} 2^{|e|})$ |
| HyperLAP | $O(log_2|\mathcal{V}| \cdot \sum_{e \in \mathcal{E}} |e|)$ | $O(|\mathcal{V}| + \sum_{e \in \mathcal{E}} |e|)$ |
| THera | $O(log_2|\mathcal{V}| \cdot \sum_{e \in \mathcal{E}} |e|)$ | $O(|\mathcal{V}| + \sum_{e \in \mathcal{E}} |e|)$ |
| **HyRec** | $O(L(|\mathcal{V}||\mathcal{E}|)^{\frac{1}{L}} + \sum_{e \in \mathcal{E}} |e|)$ | $O((|\mathcal{V}||\mathcal{E}|)^{\frac{1}{L}} + \sum_{e \in \mathcal{E}} |e|)$ |

# B ANALYSIS OF TIME AND MEMORY COMPLEXITY

Table 7 [7] presents the time and memory complexity for all models, including HyRec. The complexities of the competitors are detailed in [21], while the time and memory complexity of HyRec are formally proven in Theorem 6 and Theorem 7, respectively. For the theorems, we assume that the total Kronecker power $K$ can be evenly divided into $L$ units.

**Theorem 6** (Time Complexity of Hypergraph Generation in HyRec). *Given $L$ unit number, time complexity of hypergraph generation in HyRec is $O\left(L(|\mathcal{V}||\mathcal{E}|)^{\frac{1}{L}} + \sum_{e \in \mathcal{E}} |e|\right)$.*

**Proof.** Let the initiator matrix $\Theta \in \mathbb{R}^{N_1 \times M_1}$, where $N_1 = log_K|\mathcal{V}|$ and $M_1 = log_K|\mathcal{E}|$, represent the initiator matrix for generating a $K$-order hypergraph to reproduce $\mathcal{G} = (\mathcal{V}, \mathcal{E})$. The time complexity of computing the $k$-th power of the Kronecker product of $\Theta$ is $O(N_1^k M_1^k)$, as each element of $A$ is iterated over all elements of $B$ when computing $A \otimes B$. In addition, the time complexity of independently sampling each element is proportional to the size of the Kronecker probabilistic matrix. Under the assumption that the total Kronecker power $K$ can be evenly divided into $L$ units, the time complexity for generating the hypergraph, after dividing it into $L$ units, is $O\left(L \cdot (N_1^{\frac{K}{L}} M_1^{\frac{K}{L}})\right) = O\left(L \cdot (|\mathcal{V}|^{\frac{1}{L}} |\mathcal{E}|^{\frac{1}{L}})\right)$, reducing the computation significantly by working with smaller Kronecker powers. ___

[7]We would like to note that in all experiments, we use at least two units (i.e., $L \geq 2$).

Then, let $\hat{\Theta}^i$ represent the sampled Kronecker graph for $i$-th unit, and let $f_{NZ}(\cdot)$ represent the number of non-zero entries in the matrix. We assume that $E[f_{NZ}(\hat{\Theta}^i)] = (\sum_{e \in \mathcal{E}} |e|)^{\frac{1}{L}}$. The time complexity for computing $\bigotimes_{i=1}^{L} \hat{\Theta}^i$ depends on the number of non-zero elements in each unit $\hat{\Theta}^i$. Thus, the time complexity is $O(E[\prod_{i=1}^{L} f_{NZ}(\hat{\Theta}^i)])$. Since the sampling at each unit is independent, this can be simplified to $O(\prod_{i=1}^{L} E[f_{NZ}(\hat{\Theta}^i)]) = O(\sum_{e \in \mathcal{E}} |e|)$. Therefore, the total time complexity is $O\left(L(|\mathcal{V}||\mathcal{E}|)^{\frac{1}{L}} + \sum_{e \in \mathcal{E}} |e|\right)$. ∎

**Theorem 7** (Memory Complexity of Hypergraph Generation in HyRec). *Given $L$ unit number, memory complexity of hypergraph generation in HyRec is $O((|\mathcal{V}||\mathcal{E}|)^{\frac{1}{L}} + \sum_{e \in \mathcal{E}} |e|)$.*

**Proof.** At each unit, memory is required to compute the Kronecker power of $\Theta$, with the memory usage proportional to the size of the resulting Kronecker power. Similar to the time complexity analysis, this results in a memory complexity of $O((|\mathcal{V}||\mathcal{E}|)^{\frac{1}{L}})$. Additionally, when computing the Kronecker product of the sampled matrices from each unit, only the non-zero elements of the Kronecker graphs need to be stored, leading to a memory complexity of $O(\sum_{e \in \mathcal{E}} |e|)$. Therefore, the total memory complexity is $O((|\mathcal{V}||\mathcal{E}|)^{\frac{1}{L}} + \sum_{e \in \mathcal{E}} |e|)$. ∎

**Table 8: Time Complexity of SingFit (Algorithm 1)**

| **SingFit** (Algorithm 1) | **Time Complexity** |
|---|---|
| Generation (Line 7) | $O(L(|\mathcal{V}||\mathcal{E}|)^{\frac{1}{L}})$ |
| Computing Singular Values (Line 8) | $O(L \cdot min(|\mathcal{E}|^2|\mathcal{V}|, |\mathcal{E}||\mathcal{V}|^2)^{\frac{1}{L}})$ |
| Computing Degree Distributions (Line 9) | $O(L \cdot (\sum_{e \in \mathcal{E}} |e|)^{\frac{1}{L}} + |\mathcal{V}|)$ |
| Computing Size Distributions (Line 10) | $O(L \cdot (\sum_{e \in \mathcal{E}} |e|)^{\frac{1}{L}} + |\mathcal{E}|)$ |

SingFit is described in Algorithm 1 and implemented using PyTorch on an NVIDIA GeForce RTX 2080 Ti GPU. We analyze the time complexity line by line (Table 8):

- Line 7: The time complexity for computing the Kronecker power of $\Theta$ and sampling the binary incidence matrix is $O(L(|\mathcal{V}||\mathcal{E}|)^{\frac{1}{L}})$, as derived in the proof of Theorem 6.
- Line 8: Singular value computation is performed using the PyTorch library, which relies on the Jacobi eigenvalue solver in cuSolver. For an arbitrary matrix of size $m \times n$, where $m \geq n$, the complexity is $O(mn^2)$. Assuming the full Kronecker power is divided evenly into $L$ units, each of size $|\mathcal{V}|^{\frac{1}{L}} \times |\mathcal{E}|^{\frac{1}{L}}$, the complexity of computing singular values across all units is $O(L \cdot min(|\mathcal{E}|^2|\mathcal{V}|, |\mathcal{E}||\mathcal{V}|^2)^{\frac{1}{L}})$. To obtain the singular values for the full Kronecker power, the algorithm computes the Kronecker product of singular values from all units, which is proportional to the number of singular values of the full Kronecker power, i.e., $O(min(|\mathcal{V}|, |\mathcal{E}|))$. Therefore, the total complexity is $O(L \cdot min(|\mathcal{E}|^2|\mathcal{V}|, |\mathcal{E}||\mathcal{V}|^2)^{\frac{1}{L}} + min(|\mathcal{V}|, |\mathcal{E}|))$.

**Table 9: Properties of Hypergraphs Derived from Four Different Cases of Initiator Matrices**

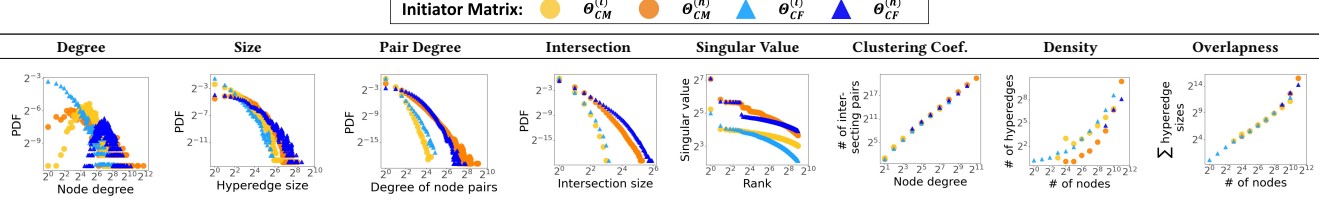

- **Line 9:** To compute the degree distribution, the algorithm first calculates the node degree vector (where the $i$-th element represents the degree of the $i$-th node) by iterating over the non-zero elements in the sampled unit matrices. Assuming $E[f_{NZ}(\hat{\Theta}^{S_i})] = (\sum_{e \in \mathcal{E}} |e|)^{\frac{1}{L}}$, where $f_{NZ}(\cdot)$ represents the number of non-zero elements in the matrix, this step has a time complexity of $O(L \cdot (\sum_{e \in \mathcal{E}} |e|)^{\frac{1}{L}})$. Next, the Kronecker product of the $L$ node degree vectors is computed, which takes $O(|\mathcal{V}|)$. Therefore, the total time complexity for computing degree distribution is $O(L \cdot (\sum_{e \in \mathcal{E}} |e|)^{\frac{1}{L}} + |\mathcal{V}|)$.

- **Line 10:** Computing the size distribution follows the same process as for the degree distribution. The Kronecker product of the size vectors corresponds to the number of edges, with a complexity of $O(|\mathcal{E}|)$. Thus, the total complexity for computing the size distribution is $O(L \cdot (\sum_{e \in \mathcal{E}} |e|)^{\frac{1}{L}} + |\mathcal{E}|)$.

## C PARAMETER SETTING

- **HyperCL** and **HyperLAP**: Both models require the distribution of node degrees and hyperedge sizes.
- **HyperFF**: $p \in [0.45, 0.48, 0.51]$ and $q \in [0.2, 0.3]$.
- **HyperPA**: It requires the distribution of hyperedge sizes and the number of new hyperedges per new node.
- **THera**: Its parameters are $C \in [8, 12, 15]$, $p \in [0.5, 0.7, 0.9]$, $\alpha \in [2, 6, 10]$, and hyperedge size distributions.
- **HyRec**: The entries of the initiator matrix are parameters. The size of the initiator incidence matrix, $N_1 \times M_1$, is determined by $N_1 = \lceil |\mathcal{V}|^{1/K} \rceil$ and $M_1 = \lceil |\mathcal{E}|^{1/K} \rceil$. Here $K$ is chosen from $k \in [2, 50]$ to minimize $\left| \left\lceil |\mathcal{V}|^{1/k} \right\rceil^k \left\lceil |\mathcal{E}|^{1/k} \right\rceil^k - |\mathcal{V}||\mathcal{E}| \right|$, subject to $1 < \left\lceil |\mathcal{V}|^{1/k} \right\rceil \left\lceil |\mathcal{E}|^{1/k} \right\rceil \leq S$, where $S \in [50, 100, 1000]$. The other parameters are (a) the learning rate $\alpha \in [0.001, 0.003, 0.005, 0.008, 0.01]$, (b) the Gumbel-Softmax temperature $\tau = 0.0005$, (c) the number of units $L \in [2, 3, 4]$; and (d) $\lambda_d \in [0.0, 0.0001, 0.001, 0.01, 0.1]$ and $\lambda_s \in [0.0, 2.0]$, which are the weights for losses.

## D SIMPLE EXAMPLES OF INITIATOR MATRIX ANALYSIS

We explore how different settings of the initiator matrix influence the properties of its Kronecker hypergraph through four cases. Specifically, we compare two structures: a community structure (CM), where nodes mainly interact within the same community, and a core-fringe structure (CF), where core nodes exist and fringe nodes primarily interact with core nodes. In both structures, we explore two cases where the probabilities in the initiator matrix differ, with one being slightly lower than the other. Thus, we analyze

the following four initiator matrices: $\Theta_{CM}^{(l)}, \Theta_{CM}^{(h)}, \Theta_{CF}^{(l)}$ and $\Theta_{CF}^{(h)}$:

$$\Theta_{CM}^{(l)} = \begin{bmatrix} 0.7 & 0.7 & 0.0001 & 0.0001 \\ 0.7 & 0.7 & 0.0001 & 0.0001 \\ 0.0001 & 0.0001 & 0.7 & 0.7 \end{bmatrix}.$$

$$\Theta_{CF}^{(l)} = \begin{bmatrix} 0.7 & 0.7 & 0.7 & 0.7 \\ 0.7 & 0.5 & 0.7 & 0.0001 \\ 0.5 & 0.5 & 0.0001 & 0.0001 \end{bmatrix}.$$

$$\Theta_{CM}^{(h)} = \begin{bmatrix} 0.9 & 0.9 & 0.0001 & 0.0001 \\ 0.9 & 0.9 & 0.0001 & 0.0001 \\ 0.0001 & 0.0001 & 0.9 & 0.9 \end{bmatrix}.$$

$$\Theta_{CF}^{(h)} = \begin{bmatrix} 0.9 & 0.9 & 0.9 & 0.9 \\ 0.9 & 0.7 & 0.9 & 0.0001 \\ 0.7 & 0.7 & 0.0001 & 0.0001 \end{bmatrix}.$$

Table 9 compares properties derived from four different initiator matrices. There are clear differences between $\Theta_{CM}^{(l)}$ ($\Theta_{CF}^{(l)}$) and $\Theta_{CM}^{(h)}$ ($\Theta_{CF}^{(h)}$), particularly as the probability increases. Higher probabilities lead to higher node degrees, larger hyperedge sizes, greater node pair degrees, and larger hyperedge intersections, but lower egonet density. When comparing the community and core-fringe structures (i.e., $\Theta_{CM}^{(h)}$ vs. $\Theta_{CF}^{(h)}$), notable differences are evident in the distributions of node degrees, density, and overlap. In the core-fringe structure $\Theta_{CF}^{(h)}$, most nodes with high degrees and the distribution centers around a peak with a rapid decline on both sides. By contrast, the community structure shows a more varied distribution of node degrees, with most nodes having relatively lower degrees. Regarding density and overlap, $\Theta_{CF}^{(h)}$ generates larger egonets, as indicated by a few data points on the right side of the distribution. This suggests that most nodes are highly interconnected, with core nodes acting as bridges that connect a significant portion of the hypergraph. However, when the probability decreases in the core-fringe structure (from $\Theta_{CF}^{(h)}$ to $\Theta_{CF}^{(l)}$), node degrees become more varied, and egonet sizes also vary, with most nodes having lower degrees and more diverse egonets.

## E SENSITIVITY OF HYREC

We analyze the effect of hyperparameters on the performance of HyRec as shown in Figure 4. The hyperparameters include (see Algorithm 1): (1) the number of parameters, i.e., the size of the initiator matrix ($N_1 \times M_1$), (2) the number of units ($L$), (3) $\lambda_s$, and (4) $\lambda_d$. We conduct experiments on two small real-world hypergraphs, email-Enron and contact-high, in terms of the number of nodes. For each experiment, performance is evaluated based on the average ranking derived from reproducing nine properties of the target hypergraph.

**Number of Parameters ($N_1 \times M_1$).** We observe improved performance (i.e., lower ranking) as the number of parameters increases.

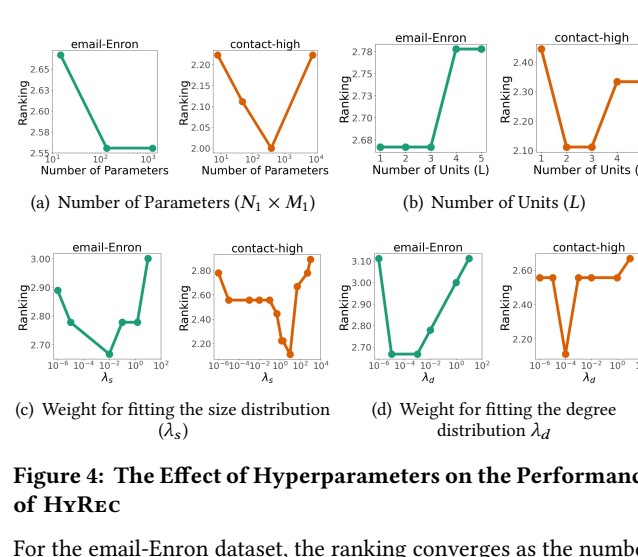

(a) Number of Parameters $(N_1 \times M_1)$   (b) Number of Units $(L)$

(c) Weight for fitting the size distribution $(\lambda_s)$   (d) Weight for fitting the degree distribution $\lambda_d$

**Figure 4: The Effect of Hyperparameters on the Performance of HyRec**

For the email-Enron dataset, the ranking converges as the number increases. For the contact-high dataset, performance improves with fewer than 1,000 parameters but declines when approaching 10,000. We hypothesize that too many parameters lead to an excess of zero entries. The difference between the space $N_1 \times M_1$ and the actual size of the hypergraph $|\mathcal{V}| \times |\mathcal{E}|$ increases as more parameters are added, introducing noise. Therefore, while increasing the number of parameters enhances performance initially, an overabundance can negatively impact the results. For this specific experiment, we fix the number of units at 2 and we explore best values for $\lambda_d$ and $\lambda_s$ within the search space detailed in Appendix C to find the best ranking for each parameter count.

**Number of Units ($L$).** Using a single unit corresponds to handling the full Kronecker power, which requires managing all possible connections between nodes and hyperedges. However, utilizing multiple units maintains the performance on the email-Enron dataset and even improves performance on the contact-high dataset, while reducing the computational complexity of HyRec, as demonstrated in the complexity analysis (Appendix B). Increasing the number of units beyond a certain point, however, does not yield further performance gains. We hypothesize this is because smaller unit sizes result in noisier singular values, reducing their usefulness. For example, when the number of units exceeds 3 in email-Enron, the smallest unit size becomes $3 \times 5$, which may generate less informative singular values compared to larger units. For this experiment, we fix the number of parameters at $3 \times 5$ and $4 \times 12$, respectively, and search for best values of $\lambda_d$ and $\lambda_s$ within the search space outlined in Appendix C, respectively, to determine the best ranking performance for each unit number.

$\lambda_s$ **and $\lambda_d$.** We observe that performance improves up to a certain point, beyond which it declines as the values of $\lambda_s$ or $\lambda_d$ exceed that point. Specifically, HyRec performs best on the email-Enron dataset when $\lambda_s = 0.01$ but performs best at $\lambda_s = 10.0$ on the contact-high dataset. Since these hyperparameters control the weights for the loss functions that match the size and degree distributions, respectively, we hypothesize that matching size distributions is more challenging for the contact-high dataset, which has a maximum hyperedge size limited to 5, thus necessitating a higher weight for $\lambda_s$. For $\lambda_d$, we observe that HyRec performs well when $\lambda_d$ is greater

than $10^{-6}$, indicating that considering degree distributions is also important. Unlike $\lambda_s$, the optimal value of $\lambda_d$ is similar for both datasets. For the specific experimental setup, we set the parameters to $(N_1, M_1, L, \lambda_s, \lambda_d) = (3, 5, 2, 0.01, 0.001)$ and $(4, 12, 2, 1.5, 0.0)$ for each dataset, which are the best hyperparameters found within the search space described in Appendix C. We then vary only the target hyperparameter $\lambda_s$ or $\lambda_d$ to evaluate the impact of each on the results.

## F LONGER-TERM EXTRAPOLATION

In this section, we extend the extrapolation task described in Section 7.3 to account for longer growth periods. The task involves forecasting hypergraph growth by fitting HyRec to an initial snapshot and predicting its future state as it expands. For this experiment, we fit HyRec to a snapshot containing hyperedges when only 1/4 of the total nodes have appeared. We then predict the properties of the full hypergraph, where the node count has quadrupled. This differs from the previous setup in Section 7.3, where HyRec was fit to 50% of the nodes. We apply the same evaluation protocol to HyperPA, HyperFF, and THera, but exclude HyperLAP and HyperCL, as they are not suitable for extrapolation beyond the input data. The results, shown in Table 10, demonstrate HyRec 's strength in predicting hypergraph evolution, especially over long-term growth. In Table 10 (a), a visual comparison in the NDC-substances dataset confirms that HyRec accurately captures both the past hypergraph and its future properties. Across 11 datasets, Table 10 (b) shows that HyRec consistently performs the **best overall** in predicting hypergraph growth, accurately modeling key properties as the node count increases fourfold. Remarkably, HyRec accomplishes this while requiring two orders of magnitude fewer input parameters (in terms of scalars) than the second-best model, THera.

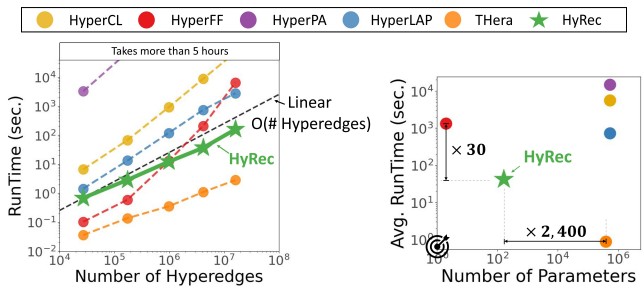

**Figure 5: Generation Time Efficiency of HyRec Compared to Other Hypergraph Generators Across Synthetic Datasets**

## G COMPARISON OF GENERATION TIME ACROSS HYPERGRAPH GENERATORS

In addition to the time complexity analysis of generation provided in Appendix B, we compare the empirical runtime of all methods, as shown in Figure 5. We use synthetic hypergraphs generated by HyRec, with an initiator matrix fitted to the email-Eu hypergraph, and vary the Kronecker power from 5 to 9 (up to $10^7$ hyperedges). Specifically, HyperFF is set with parameters $p = 0.51$ and $q = 0.3$, THera with $C = 8$, $p = 0.7$, and $\alpha = 10$, and HyRec with two units ($L = 2$). We set a runtime limit of 5 hours, so generators without markers in some cases in Figure 5 indicate that they exceeded this

**Table 10: HYREC Effectively Extrapolates Real-world Hypergraphs, Even for Extended Growth Periods.**

(a) **HYREC accurately predicts fourfold hypergraph growth:** After fitting to a past snapshot of the NDC-substances dataset with 1/4 of the total nodes, HYREC (green) accurately predicts the properties of the future snapshot (the full dataset).

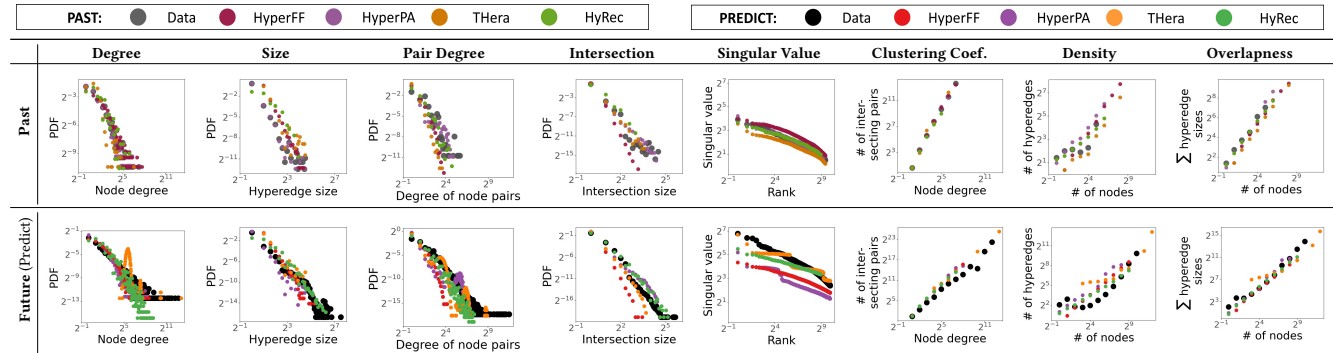

(b) **HYREC excels in extrapolation with minimal input parameters, even for long-term predictions**: HYREC performs the **best overall** in extrapolation as the node count quadruples, requiring far fewer parameters than the second-best model (THera). The best, second-best, and third-best performance are highlighted in blue, green, and yellow, respectively.

| | # Input Parameters | | Average Ranking (Across 11 Hypergraph Datasets) | | | | | | | | | |
|---|---|---|---|---|---|---|---|---|---|---|---|---|
| | **Min** | **Max** | **Degree** | **Size** | **Pair Deg.** | **Intersect.** | **Singular Value** | **Clustering Coef.** | **Density** | **Overlapness** | **Effective Diam.** | **Average** |
| HyperFF | 2 | 2 | 1.909 | 2.273 | 3.182 | 2.727 | 2.545 | 2.091 | 2.364 | 2.636 | 2.818 | 2.505 |
| HyperPA | 143 | 515,085 | 3.182 | 2.636 | 2.545 | 2.909 | 4.000 | 3.273 | 3.273 | 3.364 | 2.636 | 3.091 |
| THera | 86 | 199,803 | 2.545 | 2.000 | 2.727 | 2.000 | 1.727 | 3.182 | 2.182 | 2.455 | 2.000 | 2.313 |
| **HYREC** | 12 | 396 | 2.364 | 3.091 | 1.545 | 2.364 | 1.727 | 1.455 | 2.182 | 1.545 | 2.545 | 2.091 |

limit. Specifically, HyperFF and HyperPA surpassed the 5-hour threshold. The results show that HYREC consistently requires less time than HyperLAP, HyperCL, and HyperPA, and even for the largest hypergraph with over $10^6$ hyperedges, HYREC achieves the second-fastest runtime. While THera is the fastest method, it requires 2,400 times more input parameters (as shown on the right side of Figure 5), and HYREC outperforms it in the extrapolation task (refer to Tables 5 and 10).

# H    DISCUSSION

A limitation of HYREC is its difficulty in reproducing size distributions when the maximum hyperedge size is constrained and deviates from a heavy-tailed distribution. For instance, in the tags dataset, the number of tags assigned to a post (i.e., the size of the hyperedge) is often limited to four. Other methods (except hyperFF) can easily replicate any size distribution by generating hypergraphs directly from the input ground-truth size distribution. In such cases, HYREC may struggle to match the size distribution as precisely as these methods. However, a key strength of HYREC lies in its flexibility. Unlike other methods (again, except hyperFF) that require the hyperedge size distribution as input for each generation, HYREC operates independently of such input, allowing it to generate dynamic, evolving size distributions over time. In contrast, methods relying on static input distributions lack this adaptability, making them less effective when extrapolating to changing distributions. Although HYREC may face limitations when the distribution deviates from a heavy-tailed structure, it is important to note that many real-world hypergraphs exhibit heavy-tailed distributions, as documented in numerous studies (see Section 2.3 and our findings in Section 4). Additionally, HYREC outperforms nearly all other methods in fitting 11 real-world hypergraphs and surpasses all methods in extrapolating these hypergraphs. In contrast, HyperFF, which also does not require input size distributions, performs poorly.

For future work, we can explore extending HYREC to handle weighted, attributed, or labeled hypergraphs, which would enable it to model more complex and nuanced real-world structures. Additionally, investigating the effectiveness of the initiator matrices in capturing key features across different hypergraphs could be valuable. This could involve tasks such as graph classification, where the initiator matrices are expected to provide meaningful insights into the distinct structural properties across various domains.

