# OpenReview forum: "Kronecker Generative Models for Power Law Patterns in Real-World Hypergraphs"
_ACM.org/TheWebConf/2025/Conference — WWW 2025 Oral_

### Official Review · Reviewer_Bf7c · 2024-11-13

**Novelty:** 5
**Technical Quality:** 5

**Review:**

This paper confirms the presence of power-law characteristics in previously identified heavy-tailed distributions and introduces a new generative model, HyRec, which leverages the Kronecker product to simulate power-law distribution patterns in real-world hypergraphs. Additionally, the paper presents SingFit, a fast and memory-efficient algorithm tailored to fit the HyRec model to large-scale hypergraphs. Through mathematical proofs and empirical validation, HyRec effectively reproduces the structure and evolutionary trends of real-world hypergraphs, excelling in predicting future hypergraph growth.

The proposed methods offer several strengths. First, by verifying power-law characteristics in hypergraph attributes, this work provides a solid foundation for further exploration in this area. HyRec employs the Kronecker product as its core, extending traditional graph generation into the hypergraph domain, which enables a concise and practical approach to hypergraph generation. Furthermore, the SingFit algorithm is rigorously supported by mathematical proof, ensuring that HyRec is scalable and capable of handling large-scale datasets.

**Questions:**

Some sections in the paper, such as Section 2, are divided into very small, fragmented parts, which affects the overall conciseness and coherence. I would recommend merging related sections into fewer, broader sections to enhance logical flow and improve readability. This restructuring would provide a clearer hierarchy within the content and align better with academic writing conventions.

In Section 3.3, the definition of the log-logistic distribution lacks precision, and its relationship with the power-law distribution is not clearly explained. I would suggest using a more accurate mathematical formulation, as was done for the power-law distribution, to clarify this relationship and enhance the rigor of the section.

There is a formatting issue at the bottom of Section 4 on page 3. It would be helpful to correct this to ensure a cleaner presentation of the paper.

There appear to be multiple citation errors throughout the paper. For instance, the content cited in [27] does not align with the context in which it is referenced, and similar issues seem to occur with other citations as well. I would recommend thoroughly reviewing and correcting the citations to ensure consistency between the references and the text.

In Section 3.3, the paper discusses the relationship between the power-law distribution and the log-logistic distribution. However, in Section 4.1, when referring to Section 3.3, the paper mentions the relationship between the power-law distribution and the log-normal distribution. This is likely to cause confusion for the reader, as it seems inconsistent with the content presented in Section 3.3. I recommend clarifying this reference to avoid potential misunderstanding.

In Section 4.1, the paper discusses the heavy-tailed distributions of certain properties of hypergraphs, claiming that they follow either power-law or log-normal distributions. However, this part is not clearly presented, making it difficult to discern whether this topic has already been addressed in existing work or what specific improvements or contributions this paper makes in relation to the existing literature. I recommend providing a clearer comparison with prior work and highlighting the novel aspects of this contribution.

The Kronecker-based graph generation model is a well-established and classic approach. The HyRec model proposed in this paper builds on this foundation to apply graph generation to hypergraphs, but I find its novelty and contribution to be limited. Additionally, the properties of the singular values of the Kronecker product have been extensively studied in previous work, meaning that the theoretical basis for the SingFit algorithm is already established. As a result, I believe the innovation and contribution of this algorithm are also insufficient. Furthermore, the claim of a fast and efficient algorithm for fitting large-scale hypergraphs does not appear to offer general applicability.

**Reviewer Confidence:**

3: The reviewer is confident but not certain that the evaluation is correct

**Scope:**

3: The work is somewhat relevant to the Web and to the track, and is of narrow interest to a sub-community

---

### Official Review · Reviewer_njdc · 2024-11-18

**Novelty:** 3
**Technical Quality:** 2

**Review:**

This paper presents a novel hypergraph generative model called HyRec. The model is designed to replicate power-law and log-logistic distributions observed in various real-world hypergraphs. By leveraging the Kronecker product, HyRec generates realistic hypergraphs with a limited number of parameters. To address scalability and fitting challenges, the authors propose SingFit, an efficient algorithm for parameter estimation. Extensive experiments demonstrate that HyRec can accurately fit and extrapolate hypergraph structures, using datasets from multiple domains such as email communication, contact networks, and co-authorship graphs. The paper also provides mathematical proofs validating that HyRec can replicate key structural properties seen in real-world hypergraphs.

Pros:

1. The paper introduces a new hypergraph generative model, HyRec, that effectively extends the traditional Kronecker graph model to hypergraphs. This is a significant advancement, considering the growing importance of modeling complex group interactions in real-world networks.

2. The model leverages the Kronecker product to generate hypergraphs and derives theoretical proofs to validate its capability in replicating power-law and log-logistic distributions. This strong theoretical foundation enhances the interpretability and applicability of HyRec​.

Cons:

1. The HyRec model is based on the assumption that hypergraphs follow power-law or log-logistic distributions. This assumption may limit its applicability to datasets where these distributions do not hold, potentially reducing its generalizability​.

2. While HyRec is shown to simulate the dynamic evolution of hypergraphs, the underlying mechanisms driving these patterns are not thoroughly explained. More in-depth analysis is needed to understand why certain structural changes occur​.

3. Although the paper discusses existing models like HyperFF and THera, it lacks a detailed comparison of HyRec’s performance against these models in different scenarios. Further empirical comparisons could strengthen the paper by highlighting HyRec’s practical advantages.

4. The experimental section contains numerous tables and figures that may overwhelm readers. Simplifying or summarizing some of these results could make the findings easier to digest.

**Questions:**

1. Can the model handle hypergraphs with non-power-law distributions?

2. How does HyRec perform on extremely high-dimensional hypergraphs?

3. What explains HyRec’s predictive power in forecasting hypergraph evolution?

**Reviewer Confidence:**

2: The reviewer is willing to defend the evaluation, but it is likely that the reviewer did not understand parts of the paper

**Scope:**

3: The work is somewhat relevant to the Web and to the track, and is of narrow interest to a sub-community

---

### Official Review · Reviewer_2fRM · 2024-11-28

**Novelty:** 6
**Technical Quality:** 6

**Review:**

The paper uncovered eight interesting power-law-related patterns in real-world hypergraphs and proposed a novel generative method called HyRec, which leverages the Kronecker product to simulate hypergraphs with power-law properties. Here is a list of the pros and cons of the paper.

Pros:
1. The HyRec model and SingFit algorithm appear to be original contributions to the field of hypergraph modeling, offering a novel approach to generating hypergraphs that exhibit power-law patterns.
2. The paper is well-structured, with a clear abstract, introduction, methodology, results, and conclusion sections, which makes it accessible to readers.
3. The work is significant as it provides tools for generating synthetic hypergraphs that can be used for testing algorithms, simulating complex systems, and understanding the dynamics of real-world systems.
4. The paper provides a solid mathematical foundation for the HyRec model, including proof that it adheres to power-law patterns observed in real-world hypergraphs.

Cons:
1. There are some spelling mistakes in the paper, such as the title in Figure 2 (d), which needs to be checked carefully.
2. The model's focus on power-law patterns might limit its applicability to hypergraphs that do not exhibit such characteristics. More discussion on the model's performance with different types of hypergraphs would be beneficial.

In conclusion, the paper presents a significant contribution to the field of hypergraph modeling with its HyRec model and SingFit algorithm. It is theoretically sound, practically applicable, and well-supported by empirical evidence. While the paper excels in many areas, it could be further strengthened by addressing its limitations and broadening the discussion on its general applicability.

**Questions:**

Q1. How to set the order K of the Kronecker product? Please provide additional explanation.
Q2. Can HyRec methods be used on dynamic hypergraph datasets? HyRec model mainly focuses on the reappearance of static hypergraph structure, and may not be sufficient to capture the dynamic characteristics of hypergraph over time, such as the addition or deletion of nodes and hyperedges.
Q3. The HyRec model seems to rely on the assumption that the node degrees and hyperedge sizes of the hypergraph follow a log-logistic or power-law distribution. If real-world data does not strictly follow these distributions, will the model's fitting and extrapolation capabilities be affected? Can it be applied to random hypergraphs?

**Reviewer Confidence:**

4: The reviewer is certain that the evaluation is correct and very familiar with the relevant literature

**Scope:**

3: The work is somewhat relevant to the Web and to the track, and is of narrow interest to a sub-community

---

### Official Review · Reviewer_PJV5 · 2024-11-29

**Novelty:** 5
**Technical Quality:** 6

**Review:**

This paper studies a Kronecker product-based model to fit the distributions of hypergraphs.

Quality: Good

Clarity: Good

Originality: Average or good, as I am unfamiliar with Kronecker product-based models.

Significance: Above the average

Pros:

P1. The paper's presentation is good and mostly easy to follow.

P2. The proposed model is very neat, including a very common Kronecker product (line 443),  Gumbel-softmax for handling stochasticity (line 571).

P3. The performance looks pretty promising. E.g., in Table 4 (b) the proposed method achieves a very good trade off between the model complexity and the effectiveness.

Cons: I can hardly name a significant drawback of this paper. Maybe a very minor con is that even though hypergraphs are a kind of generalization of graphs, their application is not so broad in real-world applications due to their complexity. Again, it is not the major con of this paper, but it somewhat limits its impact on the community.

**Questions:**

Q1. Line 560 mentioned that there are two additional loss terms L_d and L_s but I cannot them anywhere, if I did not miss anything in the main content.

Q2. Page 3, the left column seems weird, where the main content overlaps with the footnote, and the top of that column is much higher than the right column. Is the page margin adjusted?

Q3. This is a very general suggestion. The overall organization of the main content seems very dense, and I suggest reorganizing some content into the appendix.

**Reviewer Confidence:**

2: The reviewer is willing to defend the evaluation, but it is likely that the reviewer did not understand parts of the paper

**Scope:**

3: The work is somewhat relevant to the Web and to the track, and is of narrow interest to a sub-community

---

### Official Review · Reviewer_coko · 2024-11-30

**Novelty:** 4
**Technical Quality:** 4

**Review:**

The paper proposes the hypergraph modelling problem and addresses a challenge in the domain of generative models for real-world hypergraphs. The proposed model, HyRec, leverages the Kronecker product effectively to generate hypergraphs with power-law characteristics observed in empirical datasets. The theoretical rigour, such as proofs for adherence to multinomial distributions, is a notable strength.

The paper is well-organized, with a logical flow from introducing hypergraph concepts to presenting experimental results. However, sections such as "Theoretical Characteristics of HyRec" and "SingFit: Fitting to Real-World Hypergraphs" are dense and could benefit from clearer explanations or visual aids to better convey the core methodologies.

**Pros**

1. The paper introduces a tractable generative model specifically for hypergraphs, filling a gap in existing literature.
2. The paper includes experiments across 11 datasets spanning six domains, with clear comparisons to established methods.
3. The paper balances accuracy and parameter efficiency, validated through theoretical proofs and empirical results.
4. Provides a stochastic extension and modular approach (via unit sampling), enabling flexible adaptation to diverse datasets.

**Cons**
1. Key concepts, such as Gumbel-Softmax for differentiable sampling, might be challenging for readers unfamiliar with advanced optimization techniques.
2. Although efficiency improvements are discussed, scalability for significantly larger datasets (e.g., billion-scale nodes) could be explored in greater depth.
3. Some figures and tables are densely packed with information, potentially overwhelming for readers.

**Questions:**

1. How does HyRec handle extremely large hypergraphs, particularly when the Kronecker power becomes computationally expensive? Are there alternative approaches to mitigate this beyond unit sampling?
2. Can the authors provide a more detailed discussion on how sensitive HyRec's performance is to the choice of initiator matrix size and the number of units (L)?
3. In what scenarios would HyRec fail to generalize? For instance, would hypergraphs with non-power-law characteristics require significant model adaptations?
4. How does the model's extrapolation performance change when applied to datasets with significant temporal evolution or abrupt structural changes?
5. While the authors mention open-source code, could additional examples or tutorials be provided to facilitate adoption?

**Reviewer Confidence:**

3: The reviewer is confident but not certain that the evaluation is correct

**Scope:**

3: The work is somewhat relevant to the Web and to the track, and is of narrow interest to a sub-community